# Cryo-EM structures of RAD51 assembled on nucleosomes containing a DSB site

Takuro Shioi[1,2], Suguru Hatazawa[1], Eriko Oya[3], Noriko Hosoya[4], Wataru Kobayashi[1], Mitsuo Ogasawara[1], Takehiko Kobayashi[2,3], Yoshimasa Takizawa[1] & Hitoshi Kurumizaka[1,2 ✉]

RAD51 is the central eukaryotic recombinase required for meiotic recombination and mitotic repair of double-strand DNA breaks (DSBs)[1,2]. However, the mechanism by which RAD51 functions at DSB sites in chromatin has remained elusive. Here we report the cryo-electron microscopy structures of human RAD51–nucleosome complexes, in which RAD51 forms ring and filament conformations. In the ring forms, the N-terminal lobe domains (NLDs) of RAD51 protomers are aligned on the outside of the RAD51 ring, and directly bind to the nucleosomal DNA. The nucleosomal linker DNA that contains the DSB site is recognized by the L1 and L2 loops—active centres that face the central hole of the RAD51 ring. In the filament form, the nucleosomal DNA is peeled by the RAD51 filament extension, and the NLDs of RAD51 protomers proximal to the nucleosome bind to the remaining nucleosomal DNA and histones. Mutations that affect nucleosome-binding residues of the RAD51 NLD decrease nucleosome binding, but barely affect DNA binding in vitro. Consistently, yeast Rad51 mutants with the corresponding mutations are substantially defective in DNA repair in vivo. These results reveal an unexpected function of the RAD51 NLD, and explain the mechanism by which RAD51 associates with nucleosomes, recognizes DSBs and forms the active filament in chromatin.

During meiosis, a DSB is enzymatically introduced in the genomic DNA to initiate genetic recombination[1]. By contrast, in mitotic cells, DSBs are frequently induced by ionizing radiation, DNA-damaging agents and undesired stalling of the replication machinery[2]. Homologous recombination (HR) is promoted at DSB sites and has essential roles in the meiotic genetic recombination and the mitotic recombinational repair of DSBs[3,4].

RAD51 is an evolutionarily conserved enzyme that functions in the HR pathway in both meiotic and mitotic cells, and accumulates on DSB sites in chromosomes[5–7]. During the HR process, RAD51 binds to DNA and forms a filamentous complex, in which a region of the DSB containing single-stranded DNA (ssDNA) is incorporated into the helical filament formed by the RAD51 multimer[8–10]. The RAD51–DNA complex then binds to undamaged DNA and promotes the homologous-pairing reaction, by which the ssDNA region pairs with the homologous double-stranded DNA (dsDNA) in an ATP-dependent manner[11–13].

In eukaryotes, the genomic DNA is compacted as chromatin, in which the nucleosome is the fundamental structural unit. In the nucleosome, two each of histones H2A, H2B, H3 and H4 form a histone octamer, and 145–147 base pairs of DNA continuously interact with the basic surface of this octamer[14]. Consequently, in the nucleosome, the DNA is left-handedly wrapped 1.65 times around the histone octamer, and becomes inaccessible to DNA-binding proteins. In the HR process, RAD51 somehow binds to the DNA tightly wrapped in the nucleosome, recognizes the DSB and forms an active nucleoprotein filament at the

DSB terminus in chromatin. However, the mechanism by which RAD51 promotes these steps in chromatin remains unclear.

## Structures of RAD51 bound to nucleosomes

To determine how RAD51 assembles on chromatin with a DSB terminus, we reconstituted the nucleosome with DNA containing the Widom 601 nucleosome positioning sequence[15]. The resulting nucleosome was positioned at one end of the DNA. At the other DNA end of the nucleosome, the eight-base-pair dsDNA plus a three-base 3′ ssDNA overhang, designed to mimic the dsDNA–ssDNA junction created at a DSB terminus, protruded as the linker DNA of the nucleosome (Fig. 1a and Extended Data Fig. 1a,b). Purified human RAD51 was then incubated with the nucleosome in the absence or presence of nucleotide cofactors, such as ADP, ATP or a non-hydrolysable ATP analogue, AMP-PNP, and the resulting RAD51–nucleosome complexes were separated by sucrose gradient ultracentrifugation in the presence of glutaraldehyde (GraFix) (Extended Data Figs. 2a, 3a, 4a and 5a).

The purified RAD51–nucleosome complexes were then visualized by cryo-electron microscopy (cryo-EM). The structures of the RAD51–nucleosome complexes were processed, and then subjected to a single-particle workflow in the RELION software package[16] (Extended Data Figs. 2–5). We found that RAD51 forms multiple conformations in the complex with the nucleosome, such as ring forms with eight (octameric), nine (nonameric) or ten (decameric) protomers, and a

[1]Laboratory of Chromatin Structure and Function, Institute for Quantitative Biosciences, The University of Tokyo, Tokyo, Japan. [2]Department of Biological Sciences, Graduate School of Science, The University of Tokyo, Tokyo, Japan. [3]Laboratory of Genome Regeneration, Institute for Quantitative Biosciences, The University of Tokyo, Tokyo, Japan. [4]Laboratory of Molecular Radiology, Center for Disease Biology and Integrative Medicine, Graduate School of Medicine, The University of Tokyo, Tokyo, Japan. ✉e-mail: kurumizaka@iqb.u-tokyo.ac.jp

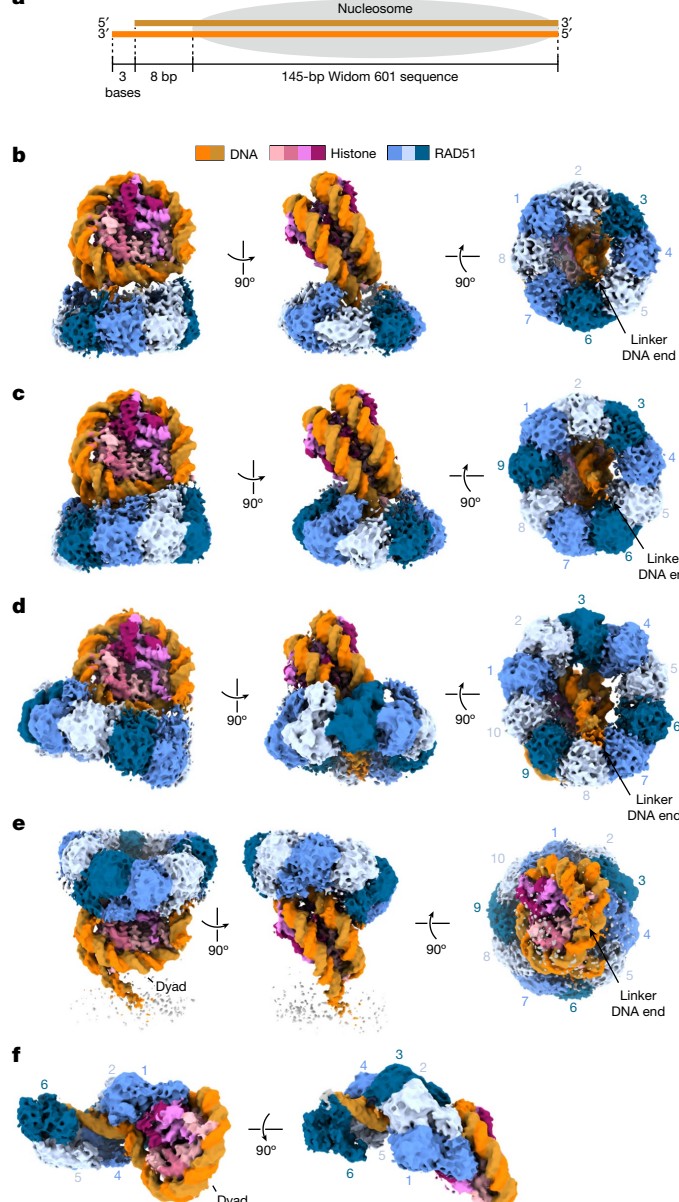

**Fig. 1 | Cryo-EM structures of the RAD51–nucleosome complexes.**
**a**, Schematic diagram of the nucleosome construct, in which one end of the DNA has an additional eight base pairs of dsDNA and a three-base ssDNA overhang. **b–e**, Three views of cryo-EM maps of the RAD51–nucleosome complexes, with the colour of each molecule corresponding to the key on the top of **b**. **b**, The octameric RAD51 ring bound to the nucleosomal DNA and linker DNA in the presence of ATP. **c**, The nonameric RAD51 ring bound to the nucleosomal DNA and linker DNA in the presence of ATP. **d**, The decameric RAD51 ring bound to the nucleosomal and linker DNA in the presence of ADP. **e**, The decameric RAD51 ring bound to the nucleosomal DNA without linker DNA binding in the presence of ATP. The dyad indicates the centre of the nucleosomal DNA. In **b–e**, the nucleosomes are aligned in the same orientation in the left, middle and right columns, respectively. The RAD51 protomers in the rings are numbered clockwise in the right column. **f**, The RAD51 filament bound to the nucleosome in the presence of AMP-PNP. The RAD51 protomers in the filament are numbered from the proximal end of the nucleosome. The dyad indicates the centre of the nucleosomal DNA.

filament form (Fig. 1b–f). The RAD51 ring forms bind to the nucleosomal DNA and incorporate the linker DNA into the central hole of the ring (Fig. 1b–d). The octameric RAD51 ring forms are found in the presence of ATP or in the absence of nucleotide cofactors. The nonameric RAD51

ring form is detected in either the presence of ATP or AMP-PNP or the absence of nucleotide cofactors. The decameric RAD51 form with the linker DNA binding is observed in the presence of ADP or AMP-PNP. Another decameric RAD51 ring form without the linker DNA binding is also detected in the presence of ADP or ATP, but not AMP-PNP (Fig. 1e). Accordingly, the decameric RAD51 ring without the bound linker DNA might represent an inactive, chromatin-associated form that serves as a standby for homeostatic DSB repair. By contrast, the RAD51 helical filament is found only in the presence of AMP-PNP, which is known to sustain the active form of RAD51 (Fig. 1f). These RAD51 rings and filament complexed with the nucleosome are separately detected on an electrophoretic mobility shift assay (Extended Data Fig. 6a). In each RAD51 ring, we designated the RAD51 protomers that first contact the nucleosomal DNA proximal to the linker DNA as 1, and numbered the successive protomers in a clockwise manner (Fig. 1b–e).

## RAD51 NLD is a nucleosome-binding module

Notably, we found that in all RAD51–nucleosome complexes, the NLDs of RAD51 directly bind to the DNA wrapped in the nucleosome (Fig. 2a–f). The NLD is not conserved in the bacterial RAD51 homologue, RecA, and has been reported to have DNA-binding activity[17] (Fig. 2a). Therefore, the RAD51 NLD might have developed evolutionarily as the nucleosome-binding module. In both the ring and the filament complexes, the Lys64 and Lys70 residues of the RAD51 NLD are located near the DNA backbone, and may directly interact with the nucleosomal DNA (Fig. 2a–f).

For the RAD51 rings with the linker DNA binding, we found that the RAD51 NLD of protomer 1 (RAD51 no. 1 NLD) consistently binds in proximity to the nucleosomal DNA entry–exit regions at the 137th base-pair position for the octameric ring, 138th for the nonameric ring and 128th for the decameric ring, from the distal end of the nucleosomal DNA (Fig. 2b–d, left). These results suggest that the RAD51 no. 1 NLD binds preferentially near the nucleosomal DNA entry–exit region across all three ring configurations. Notably, the binding mode of the RAD51 no. 1 NLDs remains consistent among these forms (Fig. 2b–d, left). The Lys64 and Lys70 residues of the NLDs seem to have a crucial role in contacting the nucleosomal DNA backbone.

Other RAD51 NLDs engage the nucleosomal DNA at various positions. The no. 6 NLD in the octameric ring, the no. 7 NLD in the nonameric ring and the no. 3 NLD in the decameric ring bind at the 77th, 72nd and 55th base-pair positions from the distal end of the DNA, respectively (Fig. 2b–d, right). These NLDs bind to the nucleosomal DNA with the binding mode observed for the RAD51 no. 1 NLD. Despite the overall structural similarity of the RAD51 promoters in the rings, a notable deviation is seen in protomer 3 of the decameric ring (with the linker DNA binding), in which the NLD orientation differs by 30° as compared with that of protomer 1, owing potentially to a unique nucleosomal DNA interaction (Fig. 2g).

In the context of the RAD51 decameric ring with the linker DNA binding, the L1 loop region of protomer 1 also contributes to the nucleosomal DNA binding, in concert with the Lys64 and Lys70 residues of the NLD (Fig. 2h). The sequential contacts at the 136th, 134th and 132nd base-pair positions from the distal end of the nucleosomal DNA by the L1 loops of RAD51 protomers 1, 2 and 3, respectively, are distinctive to the decameric ring (Fig. 2h). These interactions may not be observed in the octameric and nonameric rings, suggesting that the RAD51 decameric ring with the linker DNA binding could be an intermediate form for the structural transition from the ring to the filament.

## The RAD51 NLD interacts with histone H4

In a decameric RAD51 ring without the bound linker DNA, in addition to its nucleosomal DNA binding, an NLD of a RAD51 protomer in the RAD51 ring may also contact an N-terminal tail of histone H4 (Fig. 3a). The H4

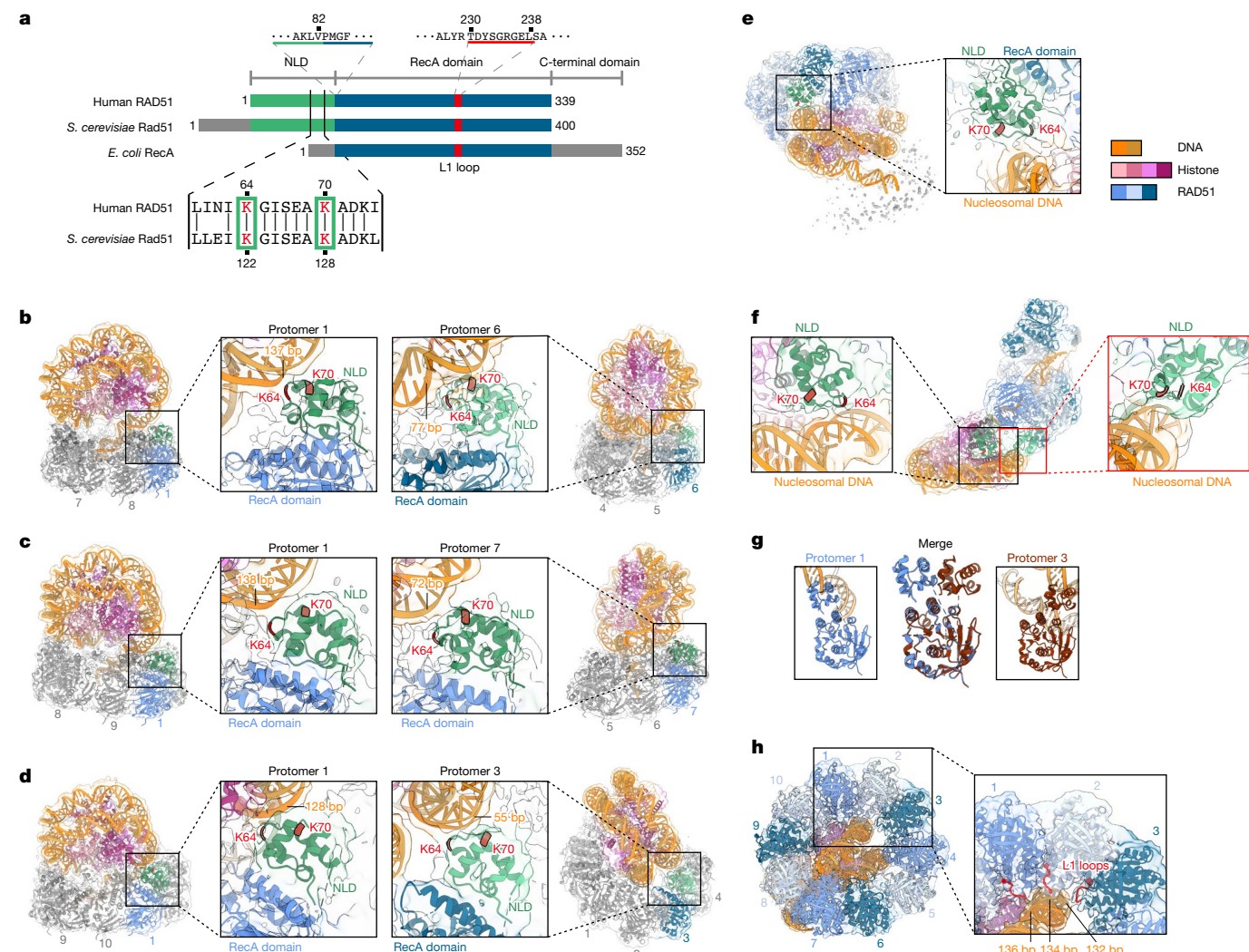

**Fig. 2 | RAD51 nucleosome binding is mediated by the NLD. a**, Comparison of the domain structures of human RAD51, *Saccharomyces cerevisiae* Rad51 and *Escherichia coli* RecA. **b**, Cryo-EM structure of the octameric RAD51 ring bound to the nucleosome with linker DNA binding (the outer sides) in the presence of ATP. Enlarged views of the RAD51 NLD–nucleosomal DNA interaction sites for RAD51 protomers 1 and 6 are shown in the middle left and middle right panels, respectively. The ChimeraX volume thresholds in the left half and right half maps are 0.00589 and 0.0041, respectively. **c**, Cryo-EM structure of the nonameric RAD51 ring bound to the nucleosome with linker DNA binding (the outer sides) in the presence of ATP, and its enlarged views (middle). **d**, Cryo-EM structure of the decameric RAD51 ring bound to the nucleosome with linker

DNA binding (the outer sides) in the presence of ADP, and its enlarged views (middle). **e**, Cryo-EM structure of the decameric RAD51 ring bound to the nucleosome, without linker DNA binding in the presence of ATP, and its enlarged view (right). **f**, Cryo-EM structure of the RAD51 filament bound to the nucleosome in the presence of AMP-PNP, and its enlarged views (left and right). **g**, Cryo-EM structures of RAD51 protomers 1 and 3 of the decameric RAD51 ring bound to the nucleosome with linker DNA binding. **h**, Cryo-EM structure of the decameric RAD51 ring bound to the linker DNA. An enlarged view of the interaction sites between the L1 loops of three RAD51 protomers (1, 2 and 3) and the nucleosomal DNA is shown on the right.

N-terminal tail extends toward a RAD51 NLD, and the H4 Lys16, Arg17, His18 and Arg19 residues are located close to the NLD (Fig. 3a). The interactions between the RAD51 NLD and the H4 tail may not be observed in the RAD51 rings bound to the linker DNA. To determine whether the H4 N-terminal tail region functions in the RAD51 ring–nucleosome interaction, we prepared nucleosome lacking the N-terminal residues 1–19 of H4 (H4 taillessΔ19; Extended Data Fig. 1a,b), and performed the RAD51 binding assay in the presence of ADP. We found that the band that corresponds to the decameric RAD51 ring in complex with the nucleosome without the bound linker DNA was specifically decreased with the taillessΔ19 H4 nucleosome (Fig. 3b and Supplementary Fig. 1a). To ascertain whether the basic Lys16, Arg17, His18 and Arg19 residues near the RAD51 NLD in the cryo-EM structure contribute to the decameric RAD51 ring–nucleosome binding, we prepared nucleosome lacking the N-terminal residues 1–15 of H4 (H4 taillessΔ15; Extended

Data Fig. 1a,b), and performed the RAD51 binding assay. As anticipated, RAD51 efficiently binds to the H4 taillessΔ15 nucleosome as well as to the wild-type nucleosome (Fig. 3c and Supplementary Fig. 1b). These results support our conclusion that the H4 N-terminal tail directly binds to an NLD of the RAD51 decameric ring. Therefore, the RAD51 ring without the bound linker DNA could be the primary nucleosome-binding form, which might function in the initial RAD51 assembly on chromatin by binding to the H4 N-terminal tail together with the nucleosomal DNA (Supplementary Video 1).

## Mutational analyses of the RAD51 NLD

To determine whether the RAD51 NLD functions in nucleosome binding, we prepared RAD51 mutants, RAD51(K64A), RAD51(K70A) and RAD51(K64A/K70A), in which Lys64, Lys70 and both residues (Lys64

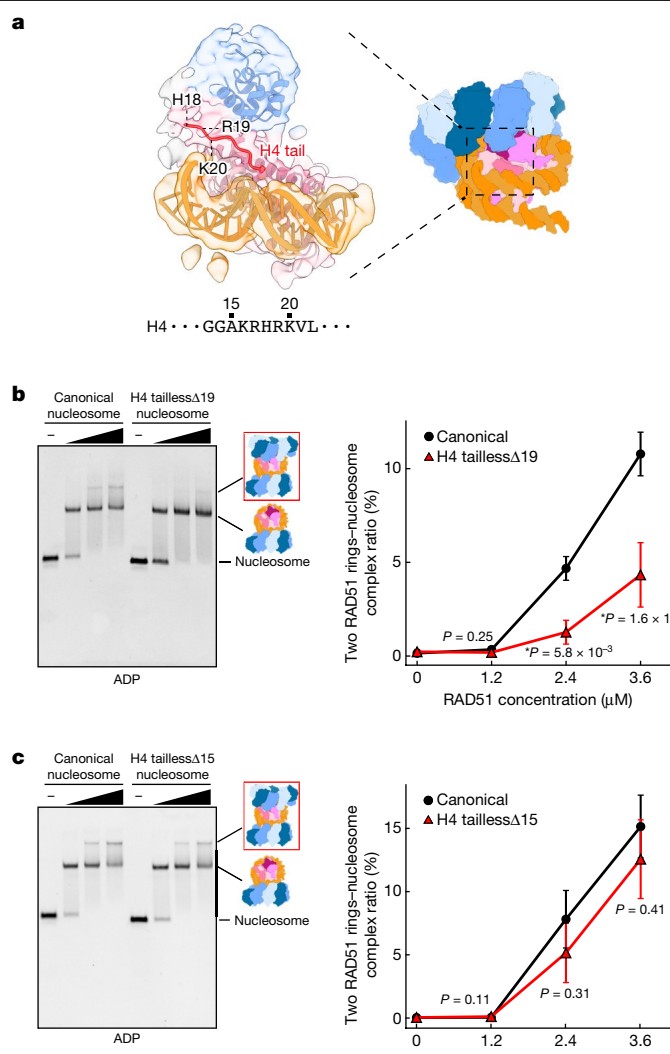

**Fig. 3 | A histone H4 N-terminal tail binds to a RAD51 NLD in the decameric RAD51 ring without linker DNA binding. a**, Focused cryo-EM structure of the histone H4 N-terminal tail bound to a RAD51 NLD. The corresponding part in the decameric RAD51 ring in complex with the nucleosome is shown on the right. **b,c**, Electrophoretic mobility shift assays of RAD51 with the canonical nucleosome and tailless Δ19 H4 nucleosome (**b**) or tailless Δ15 H4 nucleosome (**c**). The binding reaction was performed in the presence of ADP, and complex formation was analysed by non-denaturing 4% polyacrylamide gel electrophoresis with ethidium bromide staining (left). The RAD51 ring (without linker DNA binding) bound to the RAD51 ring (with linker DNA binding)–nucleosome complex is separately detected as the band that migrates more slowly than the RAD51 ring–nucleosome complex. The binding ratios of the RAD51 ring (without linker DNA binding) to the RAD51–nucleosome complex, as illustrated in the red rectangle on the right side of the gel, were estimated. The average values of three independent experiments (shown in Supplementary Fig. 1a,b) are plotted against the RAD51 concentration (right). Data are mean ± s.d. (*n* = 3 independent replicates). *P* values were obtained by two-sided Welch's *t*-test. Asterisks indicate significance at *P* < 0.05.

and Lys70), respectively, are replaced with Ala (Extended Data Fig. 1c). The RAD51 Lys64 and Lys70 residues were selected because they are conserved and in the proximity of the nucleosomal DNA. Another conserved residue, RAD51 Lys73, was not chosen because it is farther away from the nucleosomal DNA in all RAD51–nucleosome complexes. Notably, the RAD51 Lys70 mutation has been identified in cancer cells[18]. We confirmed that the RAD51 K64A and K70A mutations did not affect the filament formation activity of RAD51 by cryo-EM analysis (Extended

Data Fig. 1d). We then performed nucleosome-binding and DNA-binding assays in the presence of AMP-PNP. In terms of DNA-binding activity, the RAD51(K64A), RAD51(K70A) and RAD51(K64A/K70A) mutants are mostly proficient, albeit with slight defects (Fig. 4a and Supplementary Fig. 1c). Of note, the RAD51(K64A/K70A) mutant is markedly defective in nucleosome binding (Fig. 4b and Supplementary Fig. 1d). In addition, the RAD51(K70A) mutant exhibits a slight but clear defect in nucleosome binding (Fig. 4a,b and Supplementary Fig. 1c,d). The RAD51(R27A) mutant, in which the Arg27 residue located on the opposite NLD surface is replaced with Ala, is proficient in nucleosome binding (Extended Data Fig. 6b and Supplementary Fig. 1g). These results suggest that the RAD51 Lys64 and Lys70 residues of the NLD have a specific role in nucleosome binding in the presence of AMP-PNP.

We next tested the nucleosome-binding activity of the RAD51(K64A), RAD51(K70A) and RAD51(K64A/K70A) mutants in the presence of ADP, which mainly promotes nucleosome binding by the RAD51 ring forms (Fig. 1b–e). All three mutants are extremely defective in nucleosome binding in the presence of ADP (Fig. 4d and Supplementary Fig. 1f). Of note, although the RAD51(K70A) mutant retains DNA-binding activity, the RAD51(K64A) and RAD51(K64A/K70A) mutants are defective in DNA binding in the presence of ADP (Fig. 4c and Supplementary Fig. 1e). These results suggest that the RAD51 NLD has a role in nucleosome binding under ADP conditions, in which the ring forms of RAD51 preferentially bind the nucleosome.

The Ser67 residue of RAD51 can reportedly be phosphorylated, and its corresponding mutation in *S. cerevisiae* Rad51 is moderately defective in DNA repair in cells[19]. This suggests that phosphorylation of the RAD51 Ser67 residue enhances the formation of the active RAD51 filament. We prepared a phosphomimetic RAD51(S67E) mutant, in which the Ser67 residue is replaced with Glu (Extended Data Fig. 1c). Our nucleosome-binding assay revealed that the RAD51(S67E) mutant exhibits somewhat enhanced RAD51 filament–nucleosome complex formation, although it is substantially defective in nucleosome binding as ring forms in the presence of AMP-PNP (Extended Data Fig. 6c and Supplementary Fig. 1h). Therefore, RAD51 Ser67 phosphorylation might stimulate the conversion of RAD51 from the ring to the filament configuration in chromatin.

## Mutational analyses of the RAD51 NLD in vivo

The human RAD51 Lys64 and Lys70 residues, which are important in nucleosome binding, are conserved in the *S. cerevisiae* Rad51 as the Lys122 and Lys128 residues, respectively (Fig. 2a). To test whether the RAD51 NLD functions in DNA repair, *S. cerevisiae* Δ*rad51* strains producing Rad51 mutant proteins (Rad51(K122A), Rad51(K128A) or Rad51(K122A/K128A)) were prepared (Fig. 4e and Supplementary Fig. 2a). We then performed DNA damage sensitivity assays with these cells carrying each Rad51 mutant. For this assay, the DNA-damaging agents methyl methanesulfonate (MMS), camptothecin (CPT) and hydroxyurea (HU), which are known to be potential inducers of DSB lesions, were selected. The DNA lesions induced by these agents are reported to be at least partly repaired by the RAD51–BRCA2-mediated HR pathway[20]. We found that the yeast strain producing the Rad51(K122A/K128A) mutant is substantially defective in DNA repair (Fig. 4f, top and Supplementary Fig. 2b). In addition, the yeast cells producing the Rad51(K128A) mutant are clearly defective in DNA repair, especially in the presence of MMS, although the Rad51(K122A) mutant is only slightly defective (Fig. 4f). *S. cerevisiae* cells are extremely resistant to X-rays, as compared with mammalian cells, owing probably to an alternative repair pathway for X-ray-induced DSBs[21]. Despite this fact, the Rad51(K122A/K128A) mutant cells are clearly defective in DNA repair under the X-ray irradiation conditions (Fig. 4f, bottom and Supplementary Fig. 2b). The human RAD51(K64A), RAD51(K70A) and RAD51(K64A/K70A) mutants, which correspond to the yeast Rad51(K122A), Rad51(K128A) and Rad51(K122A/K128A) mutants, are

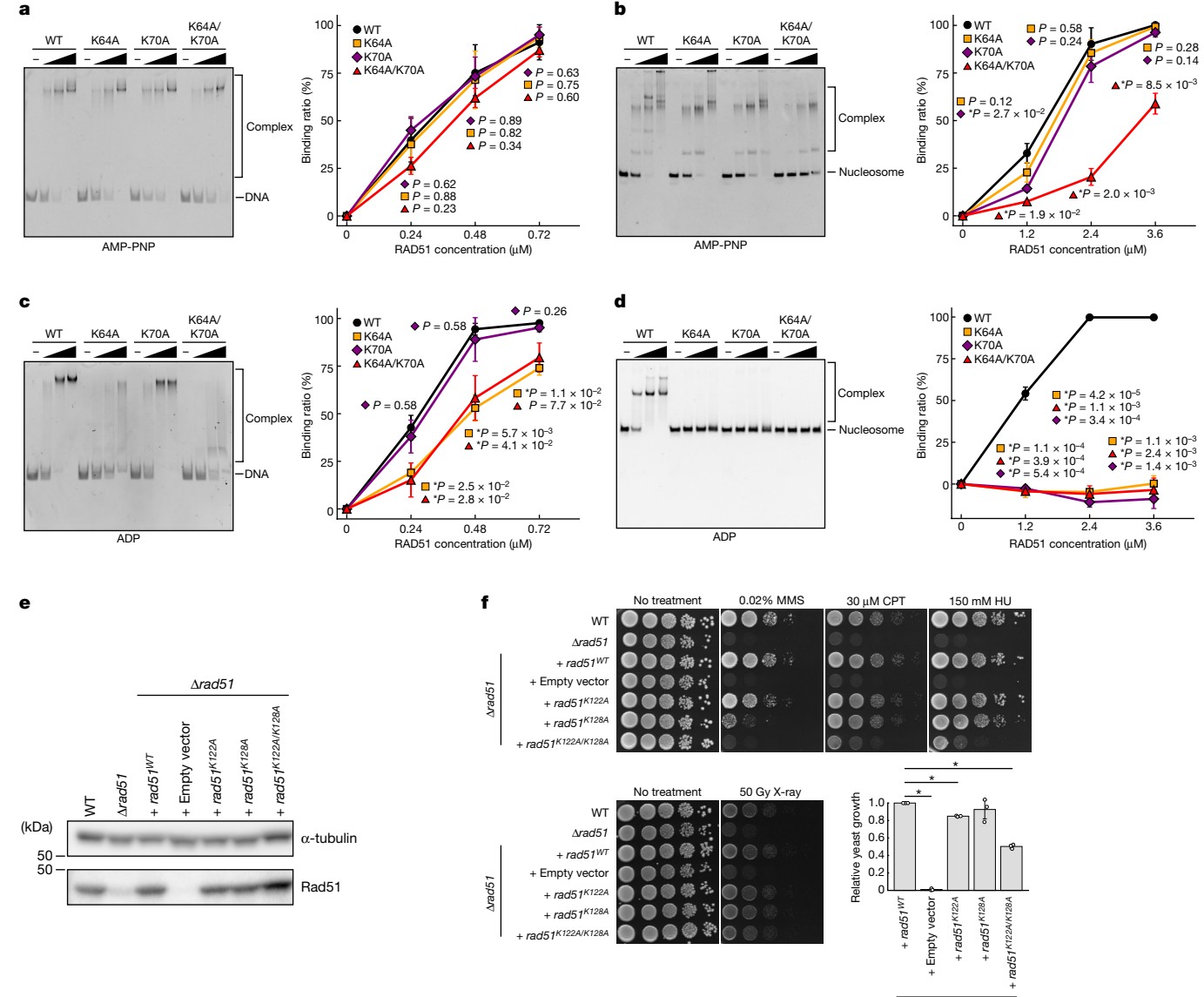

**Fig. 4 | Mutational analyses of the nucleosomal DNA-binding residues of the RAD51 NLD. a–d**, Electrophoretic mobility shift assays of RAD51, RAD51(K64A), RAD51(K70A) and RAD51(K64A/K70A) with the naked 153-bp DNA (**a**,**c**) or nucleosome (**b**,**d**). The binding reactions were conducted in the presence of AMP-PNP (**a**,**b**) or ADP (**c**,**d**). Ratios of DNA and nucleosome bound to RAD51 were estimated from the band intensities of the remaining free DNA bands (**a**,**c**, right) and free nucleosome bands (**b**,**d**, right), respectively. The average values of three independent experiments (shown in Supplementary Fig. 1c–f) are plotted against the RAD51 concentration. Data are mean ± s.d. ($n$ = 3 independent replicates). $P$ values were obtained by two-sided Welch's $t$-test. Asterisks indicate significance at $P < 0.05$. **e**, Western blot to evaluate the expression levels of RAD51 mutants, with α-tubulin as a loading control. The membrane was separated at a 50-kDa line (Supplementary Fig. 2a) and proteins were detected by each antibody. The uncropped membrane scans are shown in

Supplementary Fig. 2a. Reproducibility was confirmed by three independent experiments. WT, wild type. **f**, Spot assay for assessing the MMS, CPT, HU and X-ray sensitivities of yeast cells producing *S. cerevisiae* Rad51 mutants. Cells were spotted in tenfold serial dilutions on plates, in the absence or presence of these DNA-damaging factors. The plates were incubated at 30 °C for one day (CPT and X-ray), two days (no treatment), three days (MMS) or seven days (HU). Reproducibility of the spot assay was confirmed by three independent experiments (Supplementary Fig. 2b). Bottom right, bar graphs and dot plots showing the relative growth of yeast cells with newly introduced *Rad51* genes compared with wild-type Rad51 after X-ray irradiation. The quantification was performed using the third spot (1:100 dilution). Data are mean ± s.d. ($n$ = 3 independent replicates). $P$ values obtained by one-sided Welch's $t$-test were $1.9 \times 10^{-5}$ (+ empty vector), $2.4 \times 10^{-4}$ (+ *rad51^{K122A}*), 0.22 (+ *rad51^{K128A}*) and $3.9 \times 10^{-4}$ (+ *rad51^{K122A/K128A}*). Asterisks indicate significance at $P < 0.05$.

substantially defective in nucleosome binding (Fig. 4b,d and Supplementary Fig. 1d,f). Therefore, these results support our proposal that the RAD51 NLD functions as the nucleosome-binding module and has a key role in DNA repair in cells.

## RAD51 rings capture the nucleosomal DSB

In the first stage of the HR pathway, the DSB end is enzymatically resected by a single-strand exonuclease, generating the ssDNA tail

region[1]. The production of ssDNA by enzymatic resection has been shown to pause when the exonuclease encounters the nucleosome[22]. A block of DNA resection at the nucleosomal linker DNA might also occur through the coordinated actions of the chromatin-associating proteins[1]. These facts imply that the dsDNA–ssDNA junction could be located right next to the nucleosome (proximal linker DNA). Consistently, the linker DNA is incorporated into the central hole of the nucleosome-bound RAD51 ring (Fig. 1b–d). In this central hole, we observed the electron microscopy density of the dsDNA–ssDNA

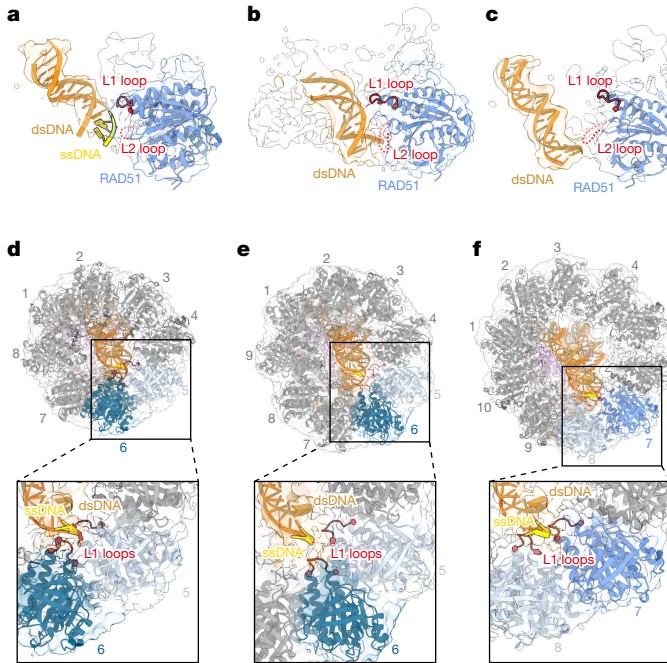

**Fig. 5 | The RAD51 L1 and L2 loops make contact near the DSB site of the linker DNA. a**, Focused cryo-EM structure of the RAD51 L1 and L2 loops bound to the sticky end of the linker DNA. **b**, Focused cryo-EM structure of RAD51 L1 and L2 bound to the blunt end of the linker DNA. **c**, Focused cryo-EM structure of the RAD51 L2 loop bound to the blunt end of the linker DNA, without the RAD51 L1 loop binding. The structures in **b**,**c** are derived from two distinct three-dimensional (3D) classes of the same sample (Extended Data Fig. 7d). **d**, The octameric RAD51 ring bound to the nucleosomal DNA and linker DNA in the presence of ATP, from the viewpoint of the linker DNA side. An enlarged view of the interaction sites between the L1 loops of RAD51 protomers (5 and 6) and the linker DNA end is shown at the bottom. **e**, The nonameric RAD51 ring bound to the nucleosomal DNA and linker DNA in the presence of ATP, from the viewpoint of the linker DNA side. An enlarged view of the interaction sites between the L1 loops of RAD51 protomers (5 and 6) and the linker DNA end is shown at the bottom. **f**, The decameric RAD51 ring bound to the nucleosomal DNA and linker DNA in the presence of ADP, from the viewpoint of the linker DNA side. An enlarged view of the interaction sites between the L1 loops of RAD51 protomers (7 and 8) and the linker DNA end is shown at the bottom.

junction region at the DSB site (Fig. 5a). Notably, this junction region is located close to a RAD51 L1 loop and may directly interact with it (Fig. 5a).

We prepared the RAD51(R235D) mutant, in which the Arg235 residue located in the L1 loop was replaced by Asp (Extended Data Fig. 1c). The RAD51(R235D) mutant is proficient in filament formation in the presence of AMP-PNP, without DNA and a nucleosome (Extended Data Fig. 1d). However, the RAD51(R235D) mutant is markedly defective in both DNA binding and nucleosome binding (Extended Data Fig. 6d,e and Supplementary Fig. 1i,j). These results suggest that, in addition to the key role of the NLD, DNA binding by the L1 loop plays an essential part in nucleosome binding, as seen in the decameric RAD51 ring bound to the nucleosome with the linker DNA (Fig. 2h).

Another RAD51 DNA-binding loop, the L2 loop, also seemed to be located near the DSB site, although its electron microscopy density is partially disordered (Fig. 5a). DNA end binding by RAD51 rings is commonly observed among the octameric, nonameric and decameric rings complexed with the nucleosome (Fig. 1b–d). The RAD51 L1 and L2 loops are known as the active DNA-binding sites for the homologous-pairing reaction[23–26]. DNA end binding by the RAD51 L1 and L2 loops is also observed in the nucleosome without the dsDNA–ssDNA junction (blunt end) in the presence of ATP (Fig. 5b and Extended Data Fig. 7).

In contrast to the DNA end binding with ssDNA, we found that in the RAD51–nucleosome complex without the ssDNA region, only the RAD51 L2 loop is located near the DNA end, whereas the L1 loop is farther from the DNA backbone (Fig. 5c). This suggests that the RAD51 L1 loop might bind preferentially to the ssDNA region around the DSB site, although it still possesses the ability to bind dsDNA. Therefore, the RAD51 ring might first assemble on the linker DNA as an active ring form, and then subsequently recognize the dsDNA–ssDNA junction by the L1 loop, when the ssDNA region that has been processed by enzymatic resection from a DSB end reaches the central hole of the RAD51 rings.

The spatial arrangements of the RAD51 octameric (or nonameric) and decameric rings relative to the nucleosome are distinct (Fig. 1b–d). Accordingly, different RAD51 protomers recognize the DNA end by the L1 loops in the central hole of the ring. In the octameric and nonameric RAD51 rings, the L1 loops of protomers 5 and 6 bind to the DNA end (Fig. 5d,e). By contrast, in the decameric RAD51 ring, the L1 loops of protomers 7 and 8 capture the DNA near the end (Fig. 5f). In addition, as shown in Fig. 2h, the L1 loops of the RAD51 decameric ring protomers 1, 2 and 3 bind to the linker DNA proximally to the nucleosome, although they might not bind directly to the DSB site. These flexible recognition mechanisms of the linker DNA containing a DSB end might allow RAD51 to bind to DSB sites in various chromatin contexts.

## The RAD51 filament is formed in the nucleosome

In the RAD51 filament complexed with the nucleosome, about 40 base pairs of the nucleosomal DNA are peeled from the histone surface by the formation of the RAD51 filament, and the DNA is sharply kinked at the DNA detachment point (Fig. 6a). The NLDs of the proximal first and second RAD51 protomers directly bind to the unpeeled nucleosomal DNA (Fig. 2f). At the DNA detachment point, the RAD51 protomer located at the proximal edge of the filament directly contacts the nucleosomal histone H2A–H2B dimer, probably through the RAD51 Glu59 residue (Fig. 6b). We prepared the RAD51(E59R) mutant, in which Glu59 was replaced by Arg (Extended Data Fig. 1c). In the presence of AMP-PNP, the RAD51(E59R) mutant exhibits clearly defective nucleosome binding by the filament form, with little effect on nucleosome binding by the ring form (Fig. 6c and Supplementary Fig. 1k). This suggests that the RAD51 NLD Glu59 residue contacts the nucleosomal H2A–H2B dimer, especially through its filament form.

The DNA region peeled from the histones is incorporated into the RAD51 filament, and may continuously interact with the RAD51 L1 loops (Fig. 6d). In this complex, six RAD51 protomers incorporating about 15 base pairs of DNA are visible in the helical filament. The DNA in the RAD51 filament is extended by about 1.5-fold, as compared with the B-form DNA. This is consistent with the previously reported active filament structure of RAD51[27,28]. About 30 base pairs of the DNA strands from the DSB terminus are not visible, which is probably due to the flexibility of this region in the complex.

## Discussion

During the meiotic HR and mitotic DSB repair processes, the DSB end is processed by nucleases, and ssDNA regions are produced by nuclease-mediated DNA resection at the DSB site[1]. In chromatin, however, DSB resection has been reported to stall when the nucleases producing the ssDNA region encounter the nucleosome[22]. Therefore, the dsDNA–ssDNA junction could be located at the linker DNA region right next to the nucleosome. RAD51 is considered to be assembled on the resulting dsDNA–ssDNA junction with the aid of the RAD51 co-activator, BRCA2[29,30]. However, the mechanism by which RAD51 finds the dsDNA–ssDNA junction point concealed by the nucleosome and forms an active filament in chromatin has been a puzzling question for many years. In this study, we present cryo-EM structures of RAD51–nucleosome complexes that reveal how RAD51 binds to the nucleosome, recognizes

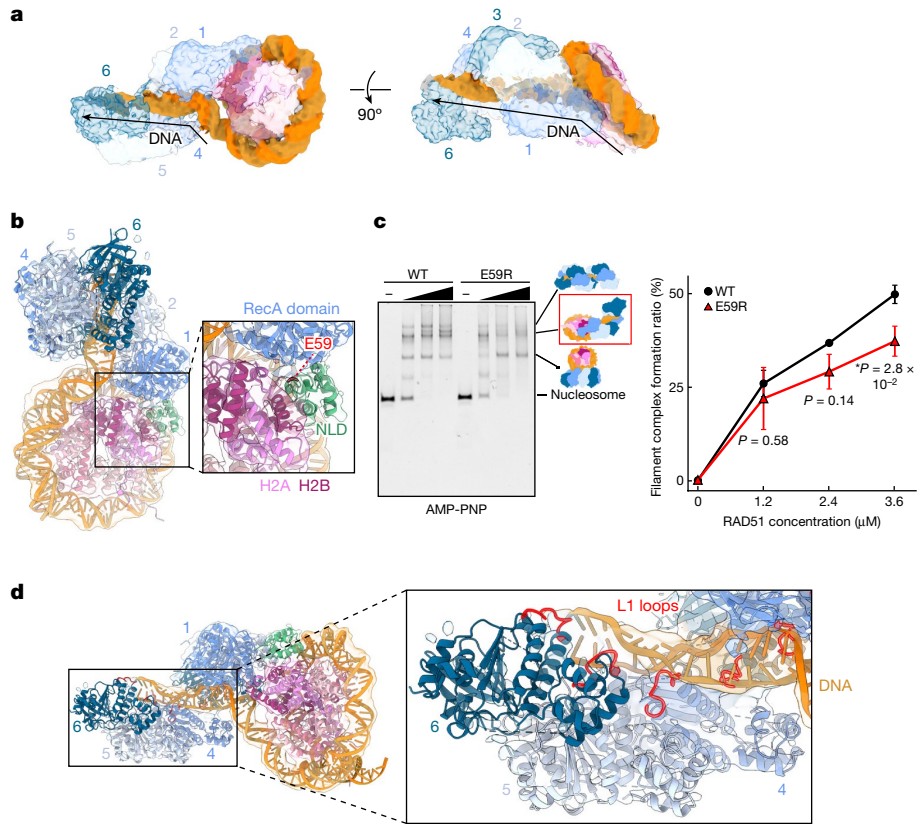

**Fig. 6 | Cryo-EM structure of the RAD51 filament bound to the nucleosome.**
**a**, Cryo-EM map of the RAD51–nucleosome complex. The DNA is coloured orange. RAD51 and histone molecules are translucent. Arrows show the DNA path in the RAD51 filament. **b**, Cryo-EM structure of the interaction sites between RAD51 and H2A–H2B in the RAD51–nucleosome complex. An enlarged view of the binding site between the nucleosomal H2A–H2B dimer and the RAD51 NLD is shown on the right. **c**, Electrophoretic mobility shift assay of RAD51 and RAD51(E59R) with the nucleosome. The binding reaction was conducted in the presence of AMP-PNP, and complex formation was analysed by non-denaturing 4% polyacrylamide gel electrophoresis with ethidium bromide staining (left). Ratios of the RAD51 filament bound to the nucleosome (red rectangle on the right side of the gel) were estimated and are plotted against the RAD51 concentration (right). The average values of three independent experiments are shown (Supplementary Fig. 1k). Data are mean ± s.d. ($n = 3$ independent replicates). $P$ values were obtained by two-sided Welch's $t$-test. Asterisks indicate significance at $P < 0.05$. **d**, Cryo-EM structure of the interaction sites between RAD51 and DNA in the RAD51–nucleosome complex. An enlarged view of the L1 loops of the RAD51 filament interacting with the extended DNA is shown on the right.

the dsDNA–ssDNA junction on the linker DNA and forms the nucleoprotein filament in chromatin.

Our cryo-EM structures of RAD51–nucleosome complexes show that the RAD51 NLD functions as the nucleosome-binding module. The NLD of RAD51 is conserved among the eukaryotic recombinases, but not in the bacterial homologue RecA[17]. Although the RAD51 NLD possesses weak DNA-binding activity, its function has not been elucidated so far[17]. We found that, in both the ring and the filament forms, RAD51 requires the NLD to bind the nucleosomal DNA (Fig. 2b–f). Our mutational analysis showed that the NLD residues Lys64 and Lys70 in human RAD51 have a specific role in nucleosome binding (Fig. 4a–d and Supplementary Fig. 1c–f). Notably, we found that yeast cells with the corresponding Rad51 NLD mutations are highly sensitive to DNA-damaging factors that potentially induce DSBs, owing probably to defective DSB repair by the HR pathway (Fig. 4f and Supplementary Fig. 2b). These facts suggest that the RAD51 NLD has an important role in DNA repair by the HR pathway in vivo, and that this is likely to be a result of its nucleosome-binding activity. The RAD51 NLD might have developed through evolution as the nucleosome-binding module in eukaryotes.

The amino acid residues on the DNA-binding surface of the RAD51 NLD are highly conserved among RAD51 proteins (Fig. 2a). Of note, the *S. cerevisiae* Rad51 Ser125 residue has been shown to be phosphorylated in the G2–M phases of the cell cycle, and this might regulate the recombination activity of Rad51[19]. In the RAD51–nucleosome complexes, the human RAD51 Ser67 residue (corresponding to the *S. cerevisiae* Rad51 Ser125 residue) is located on the binding surface of the nucleosomal DNA (Extended Data Fig. 6c and Supplementary Fig. 1h). Therefore, the phosphorylation of this site might have a major effect on the nucleosomal DNA binding of RAD51. The phosphorylation of Rad51 S125 in *S. cerevisiae* might enhance its naked DNA binding[19]. In the present study, we found that the phosphomimetic RAD51S67E mutant efficiently binds to the nucleosome in the filament form. However, this mutation reduces the nucleosome binding by the ring form of RAD51 in the presence of AMP-PNP (Extended Data Fig. 6c and Supplementary Fig. 1h). This suggests that the phosphorylation of the RAD51 Ser67 residue stimulates RAD51 to form the active filament from the ring configuration in the complexes with the nucleosome. The phosphorylation of the *S. cerevisiae* Rad51 Ser125 residue might downregulate the nucleosome binding of the RAD51 rings by repulsing the negative charge of the nucleosomal DNA phosphate backbone, and it might upregulate the DNA binding of the Rad51 filament. In fact, in the filament form, the RAD51 NLDs are located outside of the filament and might not directly contact the DNA (Fig. 6). The activation of naked DNA binding by RAD51 after its NLD phosphorylation could be coupled with the suppression of the nucleosome binding of RAD51 as ring forms. The phosphorylation of the RAD51 NLD might stimulate the conversion of the RAD51 ring to the filament, and thus enhance the active filament

formation of RAD51 on the ssDNA region at the DSB end. Future studies will be necessary to address this issue.

Mutations in human RAD51 have been found in many patients with cancer[31]. Therefore, the dysfunction of RAD51 might induce carcinogenesis or the malignant transformation of cells. In fact, RAD51 mutations in the RecA fold domain, which contains the catalytic centre for ATP hydrolysis and homologous pairing, have been found in various types of cancer cells[31]. RAD51 mutations have also been identified in the NLD region, including the K70I mutation, in which Lys70 is replaced by isoleucine[18]. Our results show that the RAD51 Lys70 residue has a key role in nucleosome binding, and its mutation induces impaired DNA repair (Fig. 4 and Supplementary Figs. 1c–f and 2). The RAD51 K70I mutation identified in cancer cells might hence cause a deficiency in DNA repair, through defective nucleosome binding, and thus promote cancer progression. Many RAD51 NLD mutations around nucleosome-binding residues—such as Glu50, Ala55, Pro56, Pro57, Ser67 and Ala69—have been identified in cancer cells[32]. The biological relevance of these NLD mutations in RAD51-dependent DNA repair will need to be studied to enable us to understand how RAD51 mutations cause carcinogenesis and malignant transformation.

In this study, we found that RAD51 binds to the nucleosome in multiple ring forms. A crystal structure of the *Pyrococcus furiosus* RAD51 ring has been reported, but its function has not been clarified so far[33]. In our cryo-EM analysis, the RAD51 NLDs aligned on the periphery of the RAD51 ring bind to the nucleosomal DNA, anchor the RAD51 ring near the linker DNA and facilitate the targeting of a dsDNA–ssDNA junction by the RAD51 L1 loop (Fig. 5). Therefore, we propose that the RAD51 rings are the predominant form for nucleosome binding. As compared to the RAD51 filament form, the RAD51 ring binds to the nucleosome without drastic structural changes of the nucleosome, such as DNA peeling from the histones (Figs. 1b–d and 2b–d). In addition, we found a decameric RAD51 ring that binds to the nucleosomal DNA but not to the linker DNA (Figs. 1e and 2e). In this complex, one of the NLDs in the RAD51 ring is in a position in which it could directly bind to the N-terminal tail of histone H4 (Fig. 3a). Deletion of the H4 N-terminal 19 residues specifically decreases the nucleosome binding by the decameric RAD51 ring (Fig. 3b and Supplementary Fig. 1a). This suggests that the H4 N-terminal tail contributes the nucleosome binding of the decameric RAD51 ring, without the bound linker DNA. Post-translational modifications of the H4 N-terminal tail reportedly affect the interactions of RAD51 with chromatin[34]. These facts imply that RAD51 is recruited on chromatin mainly as a ring form with the H4 tail, and binds to the nucleosomal DNA but not to the linker DNA; it then changes its binding position near the linker DNA to recognize the dsDNA–ssDNA junction (Supplementary Video 1). BRCA2 reportedly possesses RAD51-loading activity on the dsDNA–ssDNA junction[29,30]. This BRCA2 activity might be required in the repositioning of RAD51 to recognize the dsDNA–ssDNA junction in chromatin. Further studies are needed to investigate this point.

We also determined the structure of the RAD51 filament in complex with the nucleosome (Figs. 1f and 2f). To form the RAD51 filament on the nucleosome, the RAD51 ring bound at the dsDNA–ssDNA junction may convert into the helical filament, and about 40 base pairs of the nucleosomal DNA are peeled from the histone surface, probably by the RAD51 filament extension (Fig. 6 and Supplementary Video 1). Further filament extension by adding RAD51 protomers at its proximal side might expand the peeled region of the nucleosomal DNA, and eventually displace the histones from the DNA. Consistent with this idea, we observed many RAD51 filaments formed on DNA without nucleosomes in the presence of AMP-PNP, probably as a consequence of the complete nucleosome disassembly by the RAD51 filament extension (Extended Data Fig. 5f). This model is in good agreement with previous biochemical and biophysical results, which showed that the nucleosome is disrupted by expansion of the RAD51 filament[35–37]. The RAD51 filament could also expand on its distal side from the nucleosome, and cover

the ssDNA region produced at a DSB terminus[1,2]. It will be intriguing to study the processes through which the conversion of RAD51 from the ring to the filament form takes place at the dsDNA–ssDNA junction, to form the active RAD51 filament on the region of ssDNA that is produced at the DSB end.

In our study, we have presented snapshot structures of RAD51 multimers, and explained how RAD51 binds the nucleosome, finds the linker DNA containing the dsDNA–ssDNA junction and forms an active filament in chromatin. We found that the RAD51 NLD is important for nucleosome binding and DNA repair in cells. The RAD51 NLD might have co-evolved with the nucleosome in eukaryotes. Our structures have uncovered a new aspect of the nucleosome—its role as a functional unit in the HR process promoted by RAD51.

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

## Methods

### Purification of RAD51 proteins

The human RAD51 and RAD51 mutant proteins were purified as previously described[23]. In brief, His₆-tagged human RAD51 and RAD51 mutants (K64A, K70A, K64A/K70A, R27A, S67E, R235D and E59R) were produced in *E. coli* cells and purified by Ni-NTA agarose chromatography (Qiagen). The His₆-tag portion was removed by thrombin protease treatment. The RAD51 proteins were then precipitated with spermidine, and dissolved in potassium phosphate buffer. RAD51 proteins were further purified by MonoQ column chromatography (Cytiva).

### Purification of histones

Human histones were purified as previously described[38]. In brief, human histones (H2A, H2B, H3.1, H4, taillessΔ15 H4 and taillessΔ19 H4) were produced in *E. coli* cells as His₆-tagged peptides. His₆-tagged histones were denatured with urea, and purified by Ni-NTA agarose chromatography (Qiagen). The His₆-tag portion was removed by thrombin protease treatment, and the histones were further purified by MonoS column chromatography (Cytiva).

### Nucleosome preparation

The nucleosomes with the 153-base-pair (with a three-base 3′ ssDNA overhang) and 158-base-pair (with blunt ends) Widom 601 DNA were prepared as previously described[38,39]. In brief, the histone octamer was reconstituted with histones H2A, H2B, H3.1 and H4, and the resulting histone complex was purified by gel filtration chromatography on a HiLoad16/60 Superdex 200 column (Cytiva). The 158-base-pair Widom 601 DNA fragment with blunt ends was amplified by PCR and purified by native polyacrylamide gel electrophoresis, using a Prep Cell apparatus (Bio-Rad). The sequence of the 158-base-pair DNA fragment is as follows: 5′-CGTGGTGGCCGTTTTCGTTGTTTTTTTCT GTCTCGTGCCTGGTGTCTTGGGTGTAATCCCCTTGGCGGTTAAAACGCG GGGGACAGCGCGTACGTGCGTTTAAGCGGTGCTAGAGCTGTCTACGA CCAATTGAGCGGCCTCGGCACCGGGATTCTGAT-3′. Nucleosomes were reconstituted by the salt dialysis method and subsequently purified with a Prep Cell apparatus[38].

### Preparation of the RAD51–nucleosome complex

RAD51 (2.4 µM) and the nucleosome (0.1 µM) were mixed in reaction buffer (20 mM HEPES-NaOH (pH 7.5), 15 mM NaCl, 1 mM MgCl₂, 1 mM dithiothreitol, 0.2 mM 2-mercaptoethanol, 0.03% NP-40 and 1.5% glycerol) in the absence or presence of 1 mM nucleotide (ATP, ADP or AMP-PNP). After an incubation at 37 °C for 30 min, the resulting complexes were fixed by the GraFix method[40] in the gradient solution (15–30% sucrose and 0-0.2% glutaraldehyde gradient in 10 mM HEPES-NaOH (pH 7.5), 30 mM NaCl and 1 mM DTT). The samples were applied on top of the gradient solution and centrifuged at 27,000 rpm at 4 °C for 16 h in an SW41Ti rotor (Beckman Coulter). After the ultracentrifugation, 640-µl aliquots were obtained from the top of the gradient solution and analysed by 4% non-denaturing polyacrylamide gel electrophoresis in 0.5× TBE buffer (45 mM Tris-borate and 1 mM EDTA), followed by ethidium bromide or SYBR Gold staining. The fractions containing the RAD51–nucleosome complexes were collected, and the buffer was then exchanged using a PD-10 column (Cytiva) to the final buffer (10 mM Tris-HCl (pH 7.5), 30 mM NaCl and 1 mM dithiothreitol). The resulting sample was concentrated with an Amicon Ultra 30K filter (Merck Millipore).

### Cryo-EM grid preparation and data collection

Aliquots (2.5 µl) of the purified RAD51–nucleosome complexes in the absence or presence of nucleotide (ATP, ADP or AMP-PNP) were applied to glow-discharged Quantifoil R1.2/1.3 200-mesh Cu grids. The grids were blotted at 4 °C for 4 or 6 s under 100% humidity using a Vitrobot Mark IV (Thermo Fisher Scientific), and then rapidly frozen in liquid ethane. Cryo-EM data of the RAD51–nucleosome complexes were collected on a Krios G4 microscope (Thermo Fisher Scientific) operating at 300 kV and a magnification of 81,000× (pixel size of 1.06 Å). The data acquisition was performed using the EPU automation software (Thermo Fisher Scientific). The defocus range varied from −1.0 to −2.5 µm. A K3 BioQuantum direct electron detector (Gatan) was used, and a stack of 40 frames was obtained for each dataset. The detailed conditions that were used for obtaining the cryo-EM data are shown in Extended Data Tables 1 and 2.

### Image processing

The detailed process is shown in Extended Data Figs. 2–5 and 7. All frames in the movies of each dataset were aligned using MotionCor2[41] with dose weighting, and the contrast transfer function (CTF) estimation was then performed using CTFFIND4[42] on digital micrographs. Micrographs were selected on the basis of the strong correlation of the CTF. The following image-processing steps were performed using Relion 4 beta2[16]. Picked particles by Laplacian-of-Gaussian (LoG)-based auto-picking were subjected to two-dimensional (2D) classification, and 2D class averages with nucleosome and additional densities were used as references for the following particle picking. Picked particles were extracted from micrographs with 2× binning. Further 2D classification was performed to discard junk particles. An initial model was then generated de novo, and several rounds of 3D classification were performed using a reasonable model as a reference. After removing the 2× binning, Bayesian polishing and CTF refinement were conducted. A mask was created around the RAD51 ring, and further 3D classification was performed using the created mask. The final map was generated by using high-quality classes for sharpening in each class with various numbers of RAD51 molecules.

In the dataset of RAD51–nucleosome complexes containing the 153-base-pair DNA obtained in the presence of ATP, two classes were identified: one with the RAD51 ring bound to the linker DNA and nucleosome, and another with an additional RAD51 ring bound to the nucleosome without linker DNA binding. In the first round of 3D classification, these classes were separated. For the image processing of the RAD51 ring bound to the nucleosome without linker DNA binding, focused refinement on the RAD51-ring structure was performed. To analyse the binding of the RAD51 L1 loop to the sticky DNA end, focused refinement was performed after the CTF refinement by masking the sticky DNA end and RAD51. We conducted this focused refinement with the RAD51 protomers bound to the linker DNA without selecting specific ring structures. To analyse RAD51 binding to the histone H4 tail, focused refinement was performed by masking the region around the histone H4 tail. We conducted this focused refinement with the H4 tail without selecting specific ring structures.

In the analysis of the dataset of the RAD51–nucleosome complex containing the 158-base-pair DNA with blunt ends obtained in the presence of ATP, only the RAD51 ring bound to the linker DNA was analysed. The structures were refined separately, on the basis of the number of protomers in each RAD51 ring. To analyse the binding of the linker DNA to the RAD51 L1 loop, focused refinement was performed by masking the DNA blunt end and RAD51. We conducted this focused refinement with the RAD51 protomers bound to the linker DNA without selecting specific ring structures.

In the analysis of the samples obtained in the presence of ADP, fractions separated by GraFix were obtained: one containing complexes with RAD51 rings bound to linker DNA, and the other containing two rings of RAD51 bound to the nucleosome. For each dataset, the RAD51 ring was focused and refined, resulting in the final maps.

For the samples obtained in the presence of AMP-PNP, two datasets of the F1 and F2 fractions were collected individually (Extended Data Fig. 5). For the image processing of F1, the processes were performed as described above. For the image processing of F2, the 2D class averages of nucleosomes with additional densities were obtained, and used as the reference for particle picking. The filament structure of RAD51

bound to the nucleosome was obtained by 2D classification and two rounds of 3D classification, and used as the reference for Topaz particle picking[43]. After 2D and 3D classifications, Bayesian polishing and CTF refinement were performed without 2× binning, and the dimer and monomer structures of the nucleosome–RAD51 filament complex were obtained. In addition, 2D class averages of the naked DNA–RAD51 filament structure were obtained, and used as the reference for particle picking. By 2D classification and two rounds of 3D classification, the cryo-EM map of the naked DNA–RAD51 complex was obtained from the reference-based particle picking of the filament structure.

## Model building

The atomic models of the RAD51–nucleosome complexes were built using the atomic coordinates of the histone octamer from the human nucleosome (Protein Data Bank (PDB) ID: 5Y0C)[44] and the atomic coordinates of a 145-bp Widom 601 sequence from the *Xenopus laevis* nucleosome (PDB ID: 7OHC)[45]. The atomic model of RAD51 was built using the crystal structure of human RAD51 (PDB ID: 5NWL)[46], and refined using the cryo-EM map of the highest-resolution RAD51 single molecule with phenix.real_space_refine[47]. The atomic coordinates of the NLD and RecA domains were adjusted and fitted to each cryo-EM map. The sequences of the nucleosomal DNA and linker DNA were modified using Chimera[48]. The atomic coordinates of the DNA were refined by manual editing with ISOLDE[49] and Coot[50]. The resulting atomic coordinates of RAD51 and nucleosome were fitted to the cryo-EM map by rigid body fitting, using the 'Fit in Map' mode of ChimeraX[51]. The major clashes were modified with phenix.real_space_refine and Coot.

For model building of the histone H4 tail bound to the RAD51, the atomic coordinates were refined by manual editing with ISOLDE and Coot.

For model building of the RAD51 filament–nucleosome complex, the DNA was built by connecting the nucleosomal DNA (PDB ID: 7OHC), the kinked DNA (PDB ID: 1WD1)[52] and the extended DNA from the human RAD51 post-synaptic complex (PDB ID: 5H1C)[28]. The atomic coordinates of the nucleosomal DNA were refined by manual editing with ISOLDE.

## Assay for RAD51–nucleosome or DNA binding

The nucleosomes (0.1 μM) or the 153-base-pair DNA (0.01 μM) and RAD51 or RAD51 mutants (0.24, 0.48 and 0.72 μM for DNA-binding assay, and 1.2, 2.4 and 3.6 μM for nucleosome-binding assay) were incubated at 37 °C for 30 min in the reaction buffer (20 mM HEPES-NaOH (pH 7.5), 15 mM NaCl, 1 mM MgCl$_2$, 1 mM dithiothreitol, 0.2 mM 2-mercaptoethanol, 0.03% NP-40 and 1.5% glycerol) in the absence or presence of 1 mM ATP, ADP or AMP-PNP. The samples were analysed by 4% non-denaturing polyacrylamide gel electrophoresis in 0.5× TBE buffer (45 mM Tris-borate and 1 mM EDTA), followed by ethidium bromide staining. Band intensities were quantitated by an Amersham Imager 680 with ImageQuant TL (Cytiva).

## Visualization of RAD51 in the absence of nucleosomes and DNA

Wild-type (WT) RAD51 (92.5 μM) and the K64A/K70A (73.3 μM) and R235D (55.2 μM) mutants were incubated at 37 °C for 30 min in reaction buffer (34 mM HEPES-NaOH (pH 7.5), 135 mM NaCl, 1 mM MgCl$_2$, 0.9 mM dithiothreitol, 1.8 mM 2-mercaptoethanol, 0.03% NP-40 and 9% glycerol) in the presence of 1 mM AMP-PNP. Aliquots (2.5 μl) were applied to glow-discharged Quantifoil R1.2/1.3 200-mesh Cu grids. The grids were blotted at 4 °C for 4 or 6 s at 100% humidity, and then rapidly frozen in liquid ethane. Micrographs of RAD51 were collected on a Krios G4 microscope operated at 300 kV and a magnification of 81,000× (pixel size of 1.06 Å).

## *Saccharomyces cerevisiae* strains and DNA damage sensitivity assays

The *S. cerevisiae* strains used in this study are listed in Extended Data Table 3. The *rad51* deletion strain (Δ*rad51* strain) was generated by replacing the endogenous *Rad51* gene with the kanamycin resistance gene (*kanMX6*). To construct *rad51*Δ + *rad51* WT and mutant stains, the *rad51* deletion (*rad51*Δ*::kanMX6*) strain was transformed with DNA fragments containing *Rad51* (WT, Δ*rad51*, *rad51^K122A^*, *rad51^K128A^* or *rad51^K122A/K128A^*) *-Ura3* genes, which were amplified by PCR or purchased (Integrated DNA Technologies). Strains were selected on synthetic complete medium without uracil (SC-Uracil: 6.7 g l$^{-1}$ Difco yeast nitrogen base without amino acids (BD Biosciences, 291940), 1.92 g l$^{-1}$ yeast Synthetic Drop-out medium supplements (Merck, Y1501-20G), 2% glucose and 2% Difco Bacto Agar).

*S. cerevisiae* cells were grown at 30 °C in yeast complete medium (YPD: 1% yeast extract, 2% peptone and 2% glucose) overnight. The pre-cultures were twofold diluted in YPD medium and incubated at 30 °C for 2 h. Afterwards, $8.0 \times 10^7$ cells grown in YPD medium were collected and suspended in 1 ml of sterile water, and tenfold serial dilutions were prepared. For all spots, 5-μl aliquots of serial dilution samples were spotted on YPD plates in the absence or presence of 0.02% (v/v) MMS, 30 μM CPT or 150 mM HU. To assess the X-ray sensitivity, yeast cultures spotted onto YPD plates were irradiated with a CellRad X-ray irradiator (Faxitron Bioptics). The plates were incubated at 30 °C for several days. The quantification was performed using the third spot (1:100 dilution) of X-ray irradiation, according to a previously described method[53].

## Protein extraction from *S. cerevisiae* and western blots

Cells ($1.0 \times 10^8$) grown in YPD medium were collected and suspended in 500 μl of ice-cold sterile water, and 75 μl of lysis buffer (2 M NaOH, 7.5% 2-mercaptoethanol) was added. After an incubation on ice for 10 min, 75 μl of 50% (v/v) trichloroacetic acid was added. After another 10 min incubation on ice, pellets obtained by centrifugation were resuspended in 60 μl Laemmli Sample Buffer (Bio-Rad, 1610737) with 5% (v/v) 2-mercaptoethanol, and the pH of the suspension was adjusted to alkaline using 1 M Tris (pH 8.8). Samples were then incubated at 65 °C for 10 min and the supernatant was used as the extracted proteins.

To detect endogenous *S. cerevisiae* Rad51, the extracted proteins were separated by SDS 10%-polyacrylamide gel electrophoresis. The gels were transferred onto membranes using an iBlot 2 Gel Transfer Device (Thermo Fisher Scientific), and the membranes were blocked with Blocking One-P (Nacalai Tesque). The membranes were then probed with the rabbit anti-*S. cerevisiae* Rad51 (1:5,000; BioAcademia, 62-101) antibody, with HRP-conjugated anti-rabbit IgG (1:5,000; Merck; NA9340) as the secondary antibody. As a loading control, α-tubulin was detected by HRP-conjugated anti-tubulin α (1:5,000; Bio-Rad, MCA77P). Can Get Signal (TOYOBO) was used for antibody dilution. Signals were enhanced by ECL Prime (Cytiva) and detected using an Amersham Imager 680 (Cytiva).

## Statistical analysis

Statistical analyses were performed using R and Python. For the electrophoretic mobility shift assays, differences in band intensities were assessed between the canonical nucleosome and each nucleosome containing a histone H4 deletion mutant, as well as between RAD51 and each RAD51 mutant at each RAD51 concentration. In the spot assay, differences in spot intensities were estimated between the + WT strains and each mutant strain. Welch's *t*-test was used to assess the differences in the means of the two datasets without conducting any pre-tests, as recommended[54].

## Use of large language models

ChatGPT was used for grammatical correction of the text and supplied the basis of the Python programs. The programs were used to generate the quantification graphs and to process PDB files.

## Reporting summary

Further information on research design is available in the Nature Portfolio Reporting Summary linked to this article.

## Data availability

The cryo-EM structures and the atomic models of the RAD51–nucleosome complexes have been deposited in the Electron Microscopy Data Bank (EMDB) and the PDB, respectively. The accession codes are as follows: EMD-38228 and PDB ID 8XBT for the cryo-EM structure of the octameric RAD51 ring bound to the nucleosome with the linker DNA binding; EMD-36442 and PDB ID 8JND for the cryo-EM structure of the nonameric RAD51 ring bound to the nucleosome with the linker DNA binding; EMD-38229 and PDB ID 8XBU for the cryo-EM structure of the decameric RAD51 ring bound to the nucleosome with the linker DNA binding; EMD-36443 and PDB ID 8JNE for the cryo-EM structure of the decameric RAD51 ring bound to the nucleosome without the linker DNA binding; EMD-36444 and PDB ID 8JNF for the cryo-EM structure of the RAD51 filament bound to the nucleosome; EMD-38230 and PDB ID 8XBV for the cryo-EM structure of the RAD51 L1 and L2 loops bound to the linker DNA with the sticky end of the nucleosome; EMD-38231 and PDB ID 8XBW for the cryo-EM structure of the RAD51 N-terminal lobe domain bound to the histone H4 tail of the nucleosome; EMD-38232 and PDB ID 8XBX for the cryo-EM structure of the RAD51 L2 loop bound to the linker DNA with the blunt end of the nucleosome; and EMD-38233 and PDB ID 8XBY for the cryo-EM structure of the RAD51 L1 and L2 loops bound to the linker DNA with the blunt end of the nucleosome. Uncropped images are shown in Supplementary Figs. 2–5. Source data are provided with this paper.

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

**Acknowledgements** We are grateful to H. Ishii (Waseda University) for his contribution to the first stage of this work, and to Y. Iikura and Y. Takeda for their assistance. This work was supported in part by JSPS KAKENHI grants JP22K06098 (to Y.T.), JP23K14134 (to S.H.) and JP23H05475 (to H.K.); the Research Support Project for Life Science and Drug Discovery (BINDS) from AMED under grant JP23ama121009 (to H.K.); and JST ERATO grant JPMJER1901 (to H.K.).

**Author contributions** T.S., S.H. and W.K. prepared the RAD51–nucleosome complexes and performed biochemical analyses. T.S., S.H., M.O. and Y.T. collected cryo-EM data and performed structural analysis. T.S., E.O., N.H. and T.K. performed DNA damage sensitivity assays with *S. cerevisiae*. H.K. conceived, designed and supervised this study. T.S. prepared all figures, and T.S., S.H., Y.T. and H.K. wrote the paper. All authors discussed the results and provided comments on the manuscript.

**Competing interests** The authors declare no competing interests.

**Additional information**
**Correspondence and requests for materials** should be addressed to Hitoshi Kurumizaka.

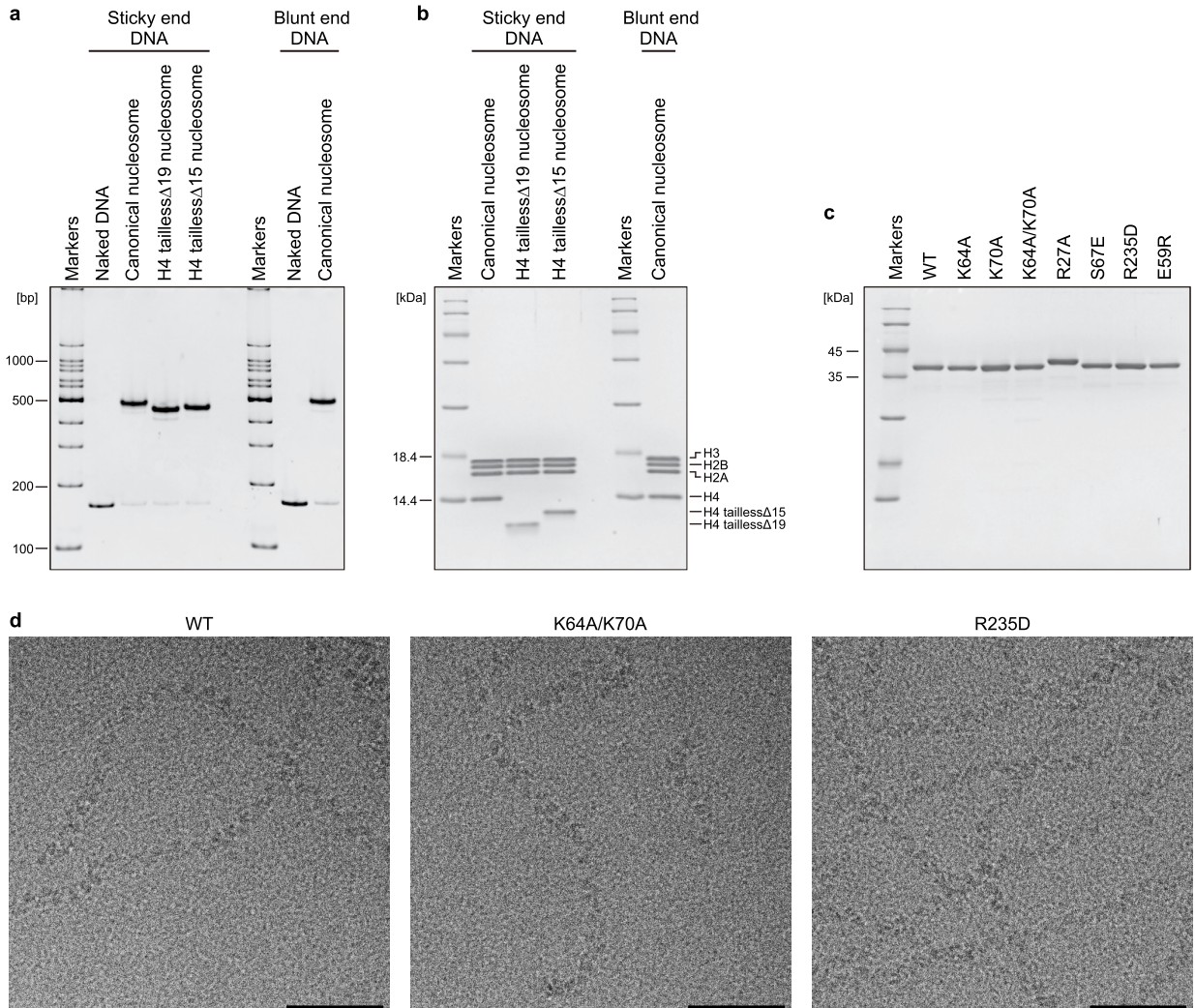

**Extended Data Fig. 1 | Preparation of the nucleosome and RAD51.**
**a**, Nucleosomes containing wild-type H4, the taillessΔ19 H4, the taillessΔ15 H4 with the sticky DNA end and the canonical nucleosome with the blunt DNA end were analysed by non-denaturing 4% polyacrylamide gel electrophoresis with ethidium bromide staining. **b**, Nucleosomes were analysed by 18% SDS polyacrylamide gel electrophoresis with Coomassie Brilliant Blue staining. **c**, Purified RAD51 and RAD51 mutants were analysed by 15% SDS polyacrylamide gel electrophoresis with Coomassie Brilliant Blue staining. **d**, Micrographs of RAD51 WT (left), RAD51(K64A/K70A) (middle) and RAD51(R235D) (right) without nucleosome and DNA in the presence of AMP-PNP. Scale bars, 50 nm.

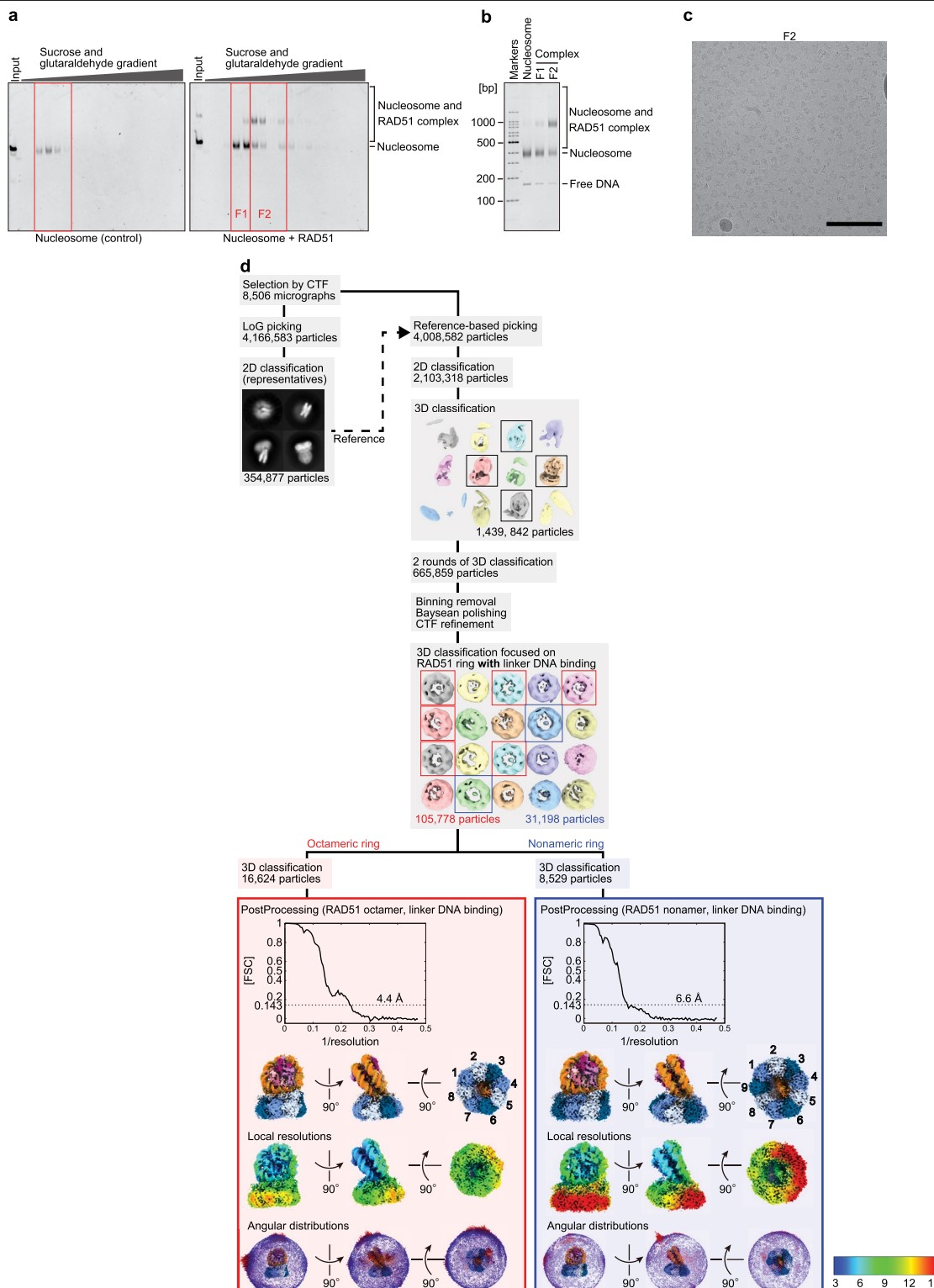

**Extended Data Fig. 2 | Structural analysis of the RAD51–nucleosome complex in the absence of nucleotide. a**, The RAD51–nucleosome complexes in the absence of nucleotide were prepared by GraFix (15–30% sucrose gradient and 0-0.2% glutaraldehyde gradient) and analysed by non-denaturing 4% polyacrylamide gel electrophoresis with ethidium bromide staining. **b**, The merged fractions enclosed by red rectangles in **a** were analysed by non-denaturing 4% polyacrylamide gel electrophoresis with ethidium bromide staining. **c**, Representative micrograph of fraction F2. Scale bar, 100 nm. **d**, Workflow of the image processing for analysing the RAD51–nucleosome complex in the absence of nucleotide, using fraction F2. The Fourier shell correlation (FSC) curves, local resolution maps and angular distributions of each map are shown in the final rectangles in each branch of the workflow. The ad-hoc low-pass filter was applied to the final maps, with a cut-off of 5.0 Å for the RAD51 octameric ring and the nonameric ring with linker DNA binding.

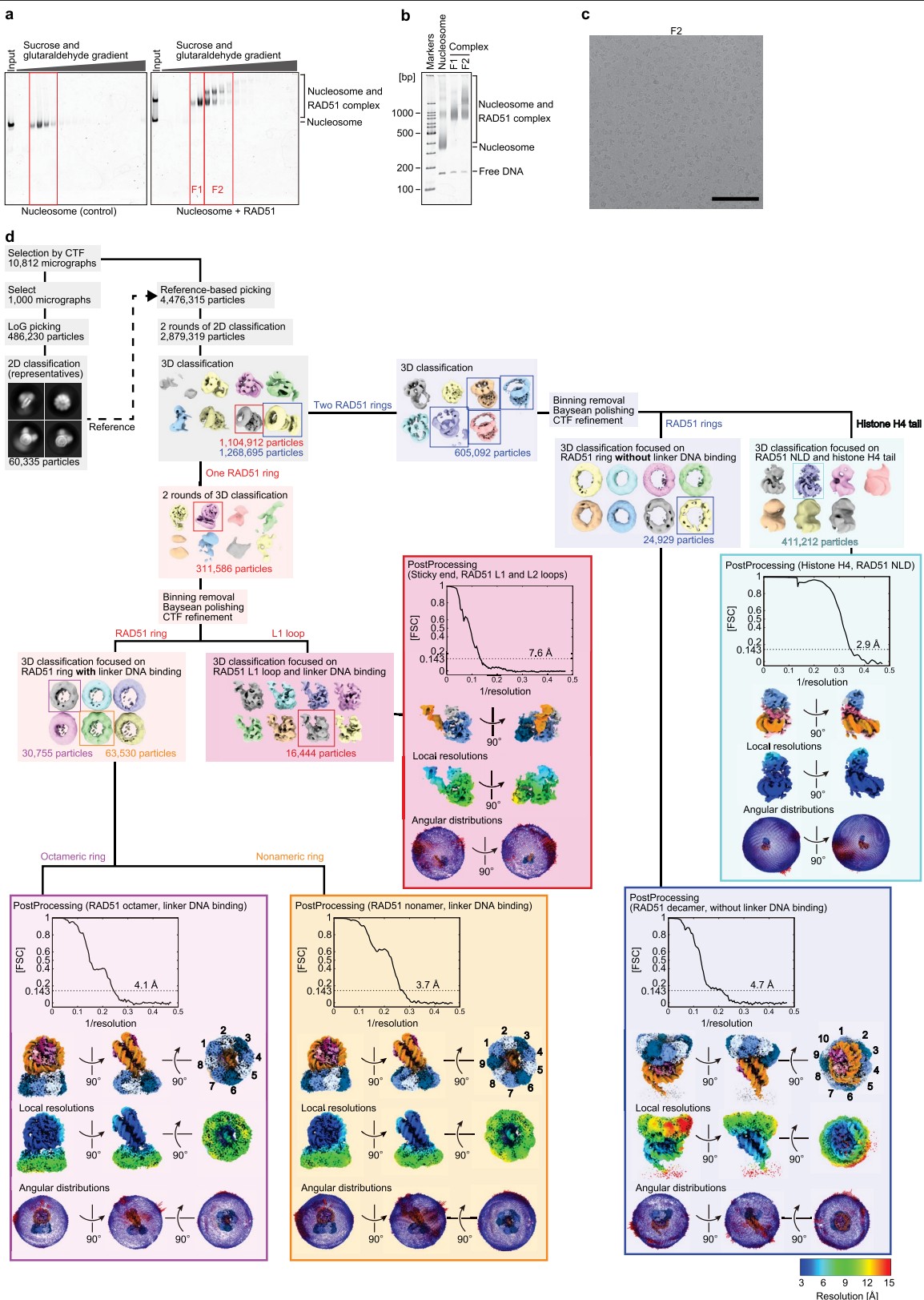

**Extended Data Fig. 3 | Structural analysis of the RAD51–nucleosome complex in the presence of ATP. a**, The RAD51–nucleosome complexes in the presence of ATP were prepared by GraFix (15–30% sucrose gradient and 0-0.2% glutaraldehyde gradient) and analysed by non-denaturing 4% polyacrylamide gel electrophoresis with SYBR Gold staining. **b**, The merged fractions enclosed by red rectangles in **a** were analysed by non-denaturing 4% polyacrylamide gel electrophoresis with SYBR Gold staining. **c**, Representative micrograph of fraction F2. Scale bar, 100 nm. **d**, Workflow of the image processing for analysing

the RAD51–nucleosome complex in the presence of ATP, using fraction F2. The Fourier shell correlation (FSC) curves, local resolution maps and angular distributions of each map are shown in the final rectangles in each branch of the workflow. The ad-hoc low-pass filter was applied to the final maps, with cut-offs of 4.7 Å for RAD51 octameric ring with linker DNA binding, RAD51 nonameric ring with linker DNA binding and RAD51 decameric ring without linker DNA binding; 5.0 Å for the focused structure of RAD51 and linker DNA binding; and 5.4 Å for the focused structure of RAD51 and histone H4 tail binding.

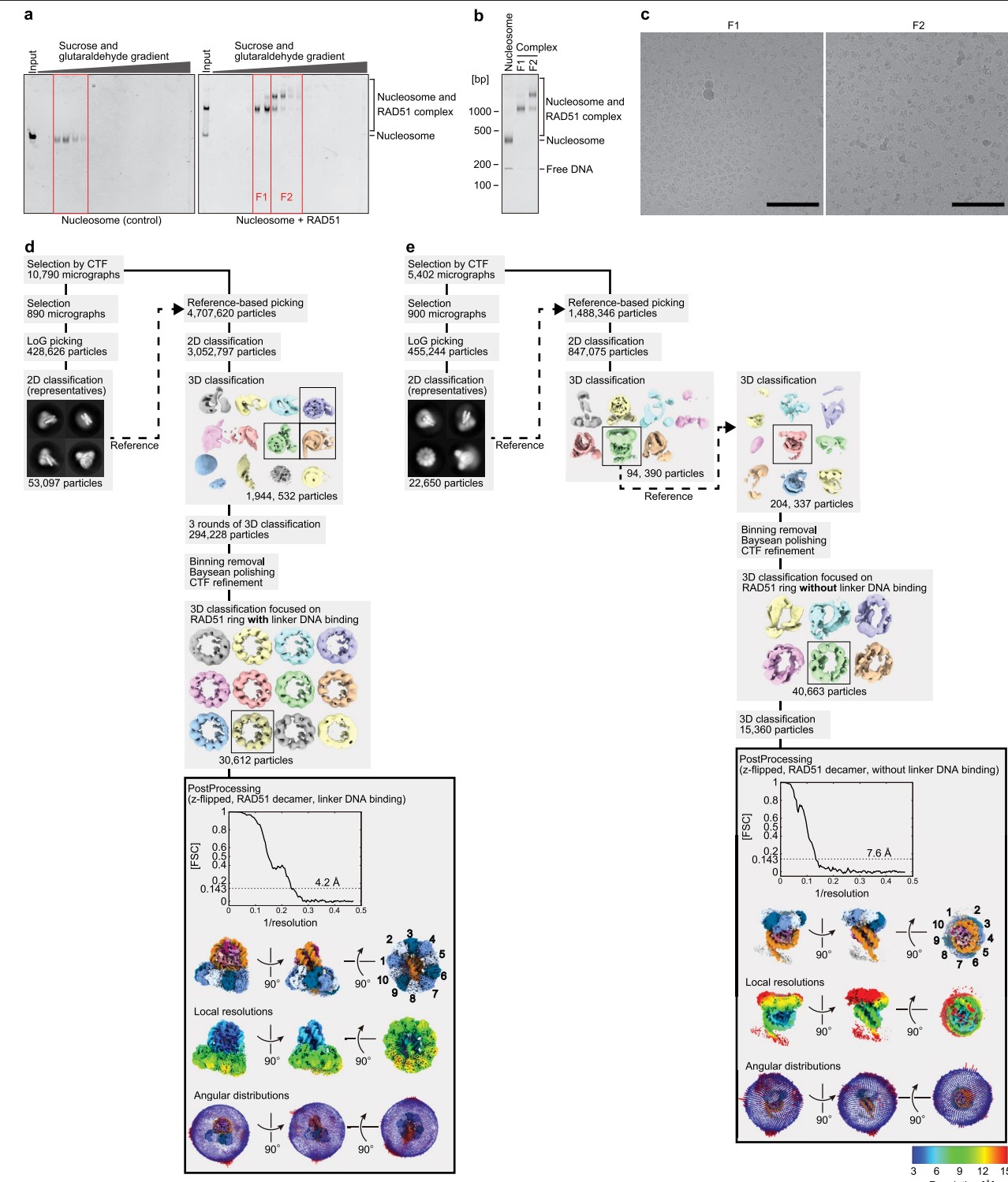

**Extended Data Fig. 4 | Structural analysis of the RAD51–nucleosome complex in the presence of ADP. a**, The RAD51–nucleosome complexes in the presence of ADP were prepared by GraFix (15–30% sucrose gradient and 0-0.2% glutaraldehyde gradient) and analysed by non-denaturing 4% polyacrylamide gel electrophoresis with ethidium bromide staining. **b**, The merged fractions enclosed by red rectangles in **a** were analysed by non-denaturing 4% polyacrylamide gel electrophoresis with ethidium bromide staining. **c**, Representative micrographs of fractions F1 and F2. Scale bars, 100 nm. **d**, Workflow of the image processing for analysing the RAD51–nucleosome complex in the presence of ADP, using fraction F1. The Fourier shell correlation (FSC) curves, local resolution maps and angular distributions of the final map are shown in the final rectangles in the workflow. The ad-hoc low-pass filter was applied to the final maps, with a cut-off of 4.7 Å for RAD51 decameric ring with linker DNA binding. **e**, Workflow of the image processing for analysing the RAD51–nucleosome complex in the presence of ADP, using fraction F2. The Fourier shell correlation (FSC) curves, local resolution maps and angular distributions of the final map are shown in the final rectangles in the workflow. The ad-hoc low-pass filter was not applied to the final maps.

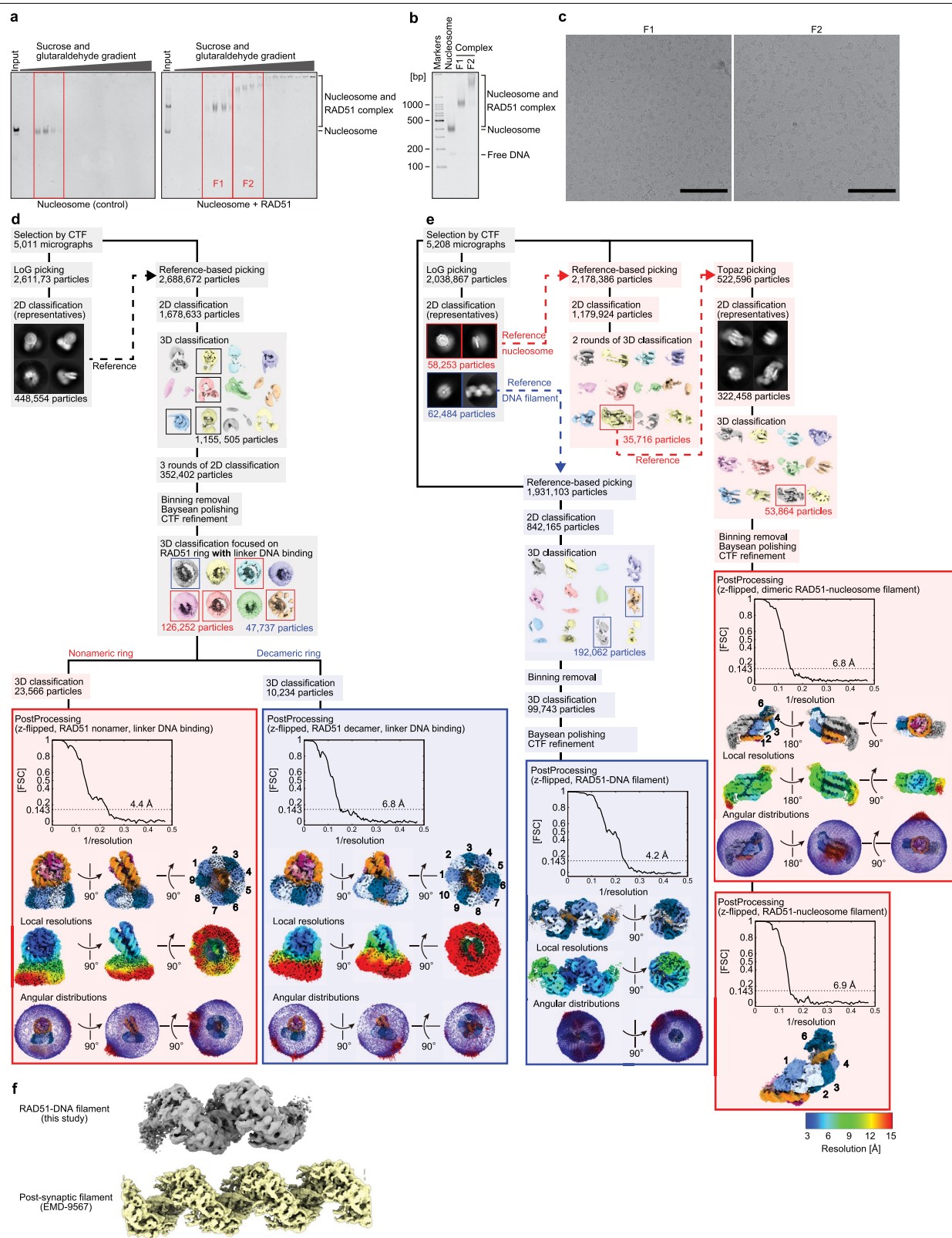

**Extended Data Fig. 5 |** See next page for caption.

**Extended Data Fig. 5 | Structural analysis of the RAD51–nucleosome complex in the presence of AMP-PNP. a**, The RAD51–nucleosome complexes in the presence of AMP-PNP were prepared by GraFix (15–30% sucrose gradient and 0-0.2% glutaraldehyde gradient) and analysed by non-denaturing 4% polyacrylamide gel electrophoresis with ethidium bromide staining. **b**, The merged fractions enclosed by red rectangles in **a** were analysed by non-denaturing 4% polyacrylamide gel electrophoresis with ethidium bromide staining. **c**, Representative micrographs of fractions F1 and F2. Scale bars, 100 nm. **d**, Workflow of the image processing for analysing the RAD51–nucleosome complex in the presence of AMP-PNP, using fraction F1. The Fourier shell correlation (FSC) curves, local resolution maps and angular distributions of each map are shown in the final rectangles in each branch of the workflow. The ad-hoc low-pass filter was applied to the final maps, with cut-offs of 4.3 Å for RAD51 nonameric ring with linker DNA binding and 6.5 Å for RAD51 decameric ring with linker DNA binding. **e**, Workflow of the image processing for analysing the RAD51–nucleosome complex in the presence of AMP-PNP, using fraction F2. The Fourier shell correlation (FSC) curves, local resolution maps and angular distributions of the final map are shown in the final rectangles in the workflow. The ad-hoc low-pass filter was applied to the final maps, with cut-offs of 4.7 Å for the RAD51–DNA filament and 5.0 Å for the dimeric and monomeric RAD51–nucleosome filament. **f**, Comparison of the RAD51–DNA filament found in this study (left) with the post-synaptic filament reported previously (right) (EMD-9567[28]).

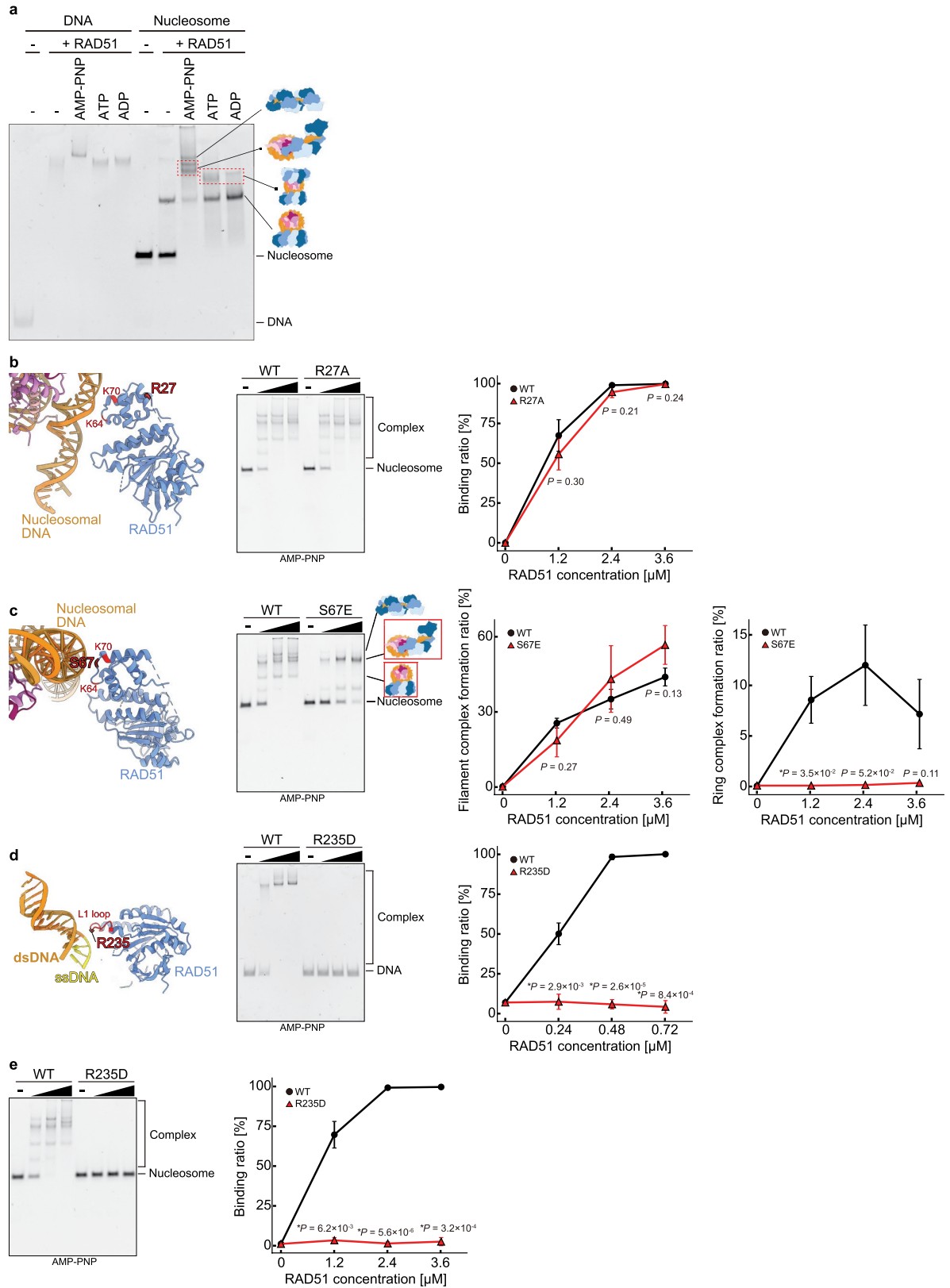

**Extended Data Fig. 6** | See next page for caption.

**Extended Data Fig. 6 | Electrophoretic mobility shift assays with RAD51 mutants. a**, The RAD51–DNA and RAD51–nucleosome complexes were analysed by non-denaturing 4% polyacrylamide gel electrophoresis with ethidium bromide staining. The bands were assigned based on cryo-EM images of the RAD51–nucleosome complexes obtained under each nucleotide condition, and the complexes are illustrated on the right side of the gel image. Reproducibility was confirmed by three independent experiments. **b**, Electrophoretic mobility shift assays of RAD51 and RAD51(R27A) with the nucleosome in the presence of AMP-PNP. The left panel shows the position of R27 in protomer 1 of the nonameric RAD51 ring bound to the nucleosomal DNA and linker DNA. The complexes were analysed by non-denaturing 4% polyacrylamide gel electrophoresis with ethidium bromide staining (middle). Ratios of the nucleosome bound to RAD51 were estimated from the band intensities of the remaining free nucleosome bands, and the average values of three independent experiments (shown in Supplementary Fig. 1g) were plotted against the RAD51 concentration (right). Data are mean ± s.d. **c**, Electrophoretic mobility shift assays of RAD51 and RAD51(S67E) in the presence of AMP-PNP. The left panel shows the position of S67 in protomer 1 of the nonameric RAD51 ring bound to the nucleosomal DNA. The RAD51 filament–nucleosome and RAD51 ring–nucleosome complexes were analysed by non-denaturing 4% polyacrylamide gel electrophoresis with ethidium bromide staining (middle left), and the ratios of these complexes were estimated (middle right and outer right). The average values of three independent experiments (shown in Supplementary Fig. 1h) were plotted against the RAD51 concentration. Data are mean ± s.d. **d**, Electrophoretic mobility shift assays of RAD51 and RAD51(R235D) with the 153 bp DNA in the presence of AMP-PNP. The left panel shows the position of R235D in the structure of RAD51 bound to the sticky linker DNA end of the nucleosome. The complexes were analysed by non-denaturing 4% polyacrylamide gel electrophoresis with ethidium bromide staining (middle). Ratios of the DNA bound to RAD51 were estimated from the band intensities of the remaining free DNA bands, and the average values of three independent experiments (shown in Supplementary Fig. 1i) were plotted against the RAD51 concentration (right). Data are mean ± s.d. **e**, Electrophoretic mobility shift assays of RAD51 and RAD51(R235D) with the nucleosome in the presence of AMP-PNP. The complexes were analysed by non-denaturing 4% polyacrylamide gel electrophoresis with ethidium bromide staining (left). Ratios of the nucleosome bound to RAD51 were estimated from the band intensities of the remaining free nucleosome bands, and the average values of three independent experiments (shown in Supplementary Fig. 1j) were plotted against the RAD51 concentration (right). Data are mean ± s.d. The $P$ values in all graphs were obtained by two-sided Welch's $t$-test. Asterisks (*) indicate significance at $P < 0.05$.

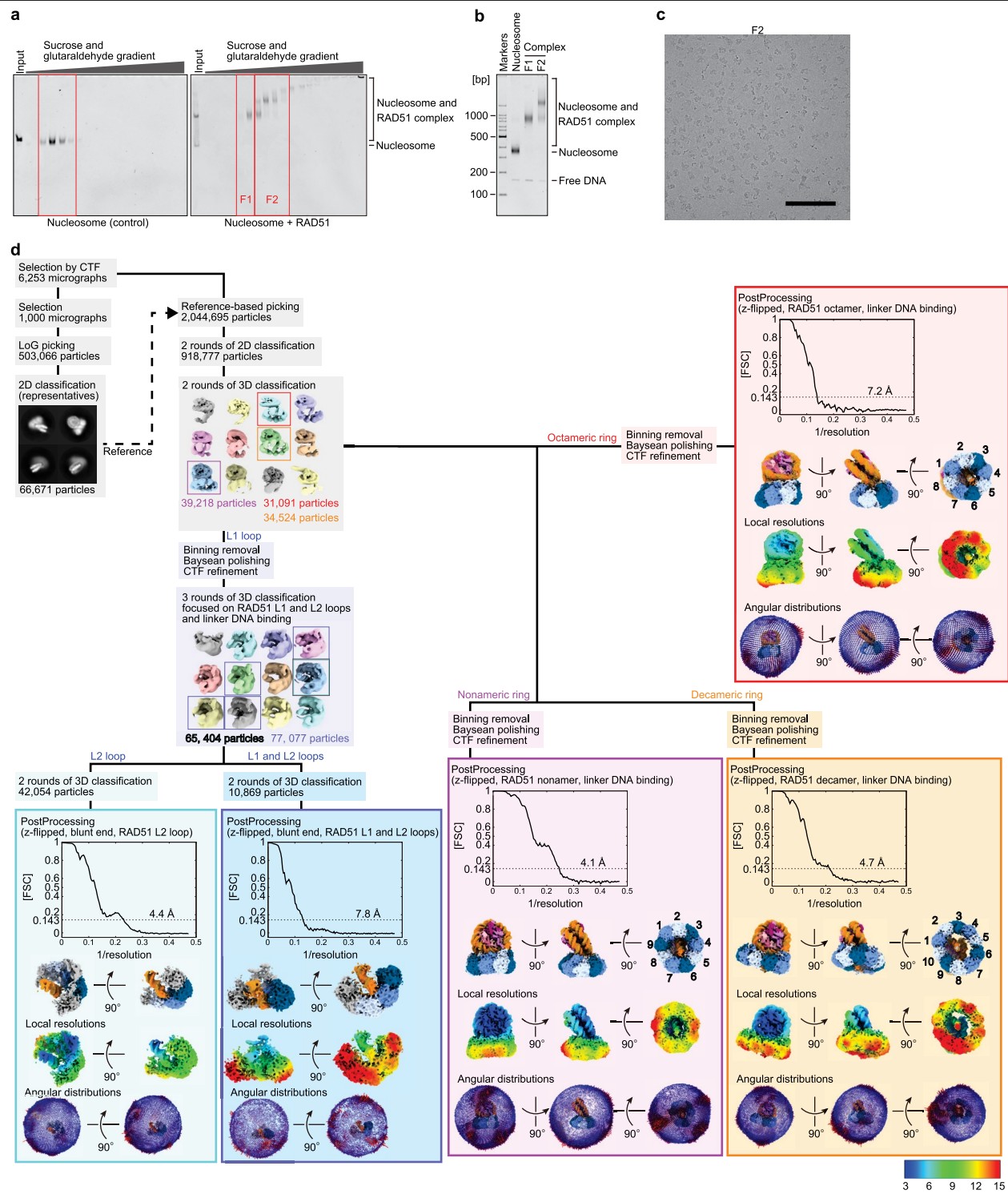

**Extended Data Fig. 7 | Structural analysis of the complex of RAD51 and the nucleosome containing 158 bp blunt-end DNA in the presence of ATP.**
**a**, The RAD51–nucleosome complexes in the presence of ATP were prepared by GraFix (15–30% sucrose gradient and 0–0.2% glutaraldehyde gradient) and analysed by non-denaturing 4% polyacrylamide gel electrophoresis with ethidium bromide staining. **b**, The merged fractions enclosed by red rectangles in **a** were analysed by non-denaturing 4% polyacrylamide gel electrophoresis with ethidium bromide staining. **c**, Representative micrograph of fraction F2. Scale bar, 100 nm. **d**, Workflow of the image processing for analysing the RAD51–

nucleosome complex in the presence of ATP, using fraction F2. The Fourier shell correlation (FSC) curves, local resolution maps and angular distributions of each map are shown in the final rectangles in each branch of the workflow. The ad-hoc low-pass filter was applied to the final maps, with cut-offs of 5.5 Å for focused structures of RAD51 L1 and L2 loops and linker DNA bindings; 5.0 Å for RAD51 octameric ring with linker DNA binding; 5.7 Å for RAD51 nonameric ring with linker DNA binding; and 6.0 Å for RAD51 decameric ring with linker DNA binding.

**Extended Data Table 1 | Cryo-EM data collection, refinement and validation statistics 1**

| | #1 The cryo-EM structure of the octameric RAD51 ring bound to the nucleosome with the linker DNA binding (EMDB-38228) (PDB 8XBT) | #2 The cryo-EM structure of the nonameric RAD51 ring bound to the nucleosome with the linker DNA binding (EMDB-36442) (PDB 8JND) | #3 The cryo-EM structure of the decameric RAD51 ring bound to the nucleosome with the linker DNA binding (EMDB-38229) (PDB 8XBU) | #4 The cryo-EM structure of the decameric RAD51 ring bound to the nucleosome without the linker DNA binding (EMDB-36443) (PDB 8JNE) |
|---|---|---|---|---|
| **Data collection and processing** | | | | |
| Magnification | 81,000× | 81,000× | 81,000× | 81,000× |
| Voltage (kV) | 300 | 300 | 300 | 300 |
| Electron exposure (e–/Å$^2$) | 1.50 | 1.50 | 1.47 | 1.50 |
| Defocus range (μm) | -1.0 to -2.5 | -1.0 to -2.5 | -1.0 to -2.5 | -1.0 to -2.5 |
| Pixel size (Å) | 1.06 | 1.06 | 1.06 | 1.06 |
| Symmetry imposed | C1 | C1 | C1 | C1 |
| Initial particle images (no.) | 4,476,315 | 4,476,315 | 4,707,620 | 4,476,315 |
| Final particle images (no.) | 30,755 | 24,929 | 30,612 | 24,929 |
| Map resolution (Å) | 4.1 | 3.7 | 4.2 | 4.7 |
| FSC threshold | 0.143 | 0.143 | 0.143 | 0.143 |
| Map resolution range (Å) | 3.3 to 16 | 3.0 to 14 | 3.4 to 14 | 3.6 to 22 |
| | | | | |
| **Refinement** | | | | |
| Initial model used (PDB code) | PDB 5Y0C, 7OHC, and 5NWL | PDB 5Y0C, 7OHC, and 5NWL | PDB 5Y0C, 7OHC, and 5NWL | PDB 5Y0C, 7OHC, and 5NWL |
| Model resolution (Å) | 4.7 | 4.6 | 4.7 | 6.9 |
| FSC threshold | 0.5 | 0.5 | 0.5 | 0.5 |
| Map sharpening *B* factor (Å$^2$) | -14.5 | -19.9 | -15.1 | -17.2 |
| Model composition | | | | |
| Non-hydrogen atoms | 27776 | 30404 | 32567 | 32956 |
| Protein residues | 2770 | 3115 | 3398 | 3452 |
| Nucleotide | 309 | 309 | 309 | 309 |
| *B* factors (Å$^2$) | | | | |
| Protein | 512.99 | 558.00 | 591.48 | 381.90 |
| Nucleotide | 250.94 | 233.11 | 187.94 | 200.93 |
| R.m.s. deviations | | | | |
| Bond lengths (Å) | 0.011 | 0.006 | 0.009 | 0.006 |
| Bond angles (°) | 1.574 | 1.049 | 1.167 | 1.219 |
| Validation | | | | |
| MolProbity score | 2.01 | 1.55 | 1.75 | 1.91 |
| Clashscore | 9.06 | 10.87 | 17.68 | 26.10 |
| Poor rotamers (%) | 2.05 | 0.04 | 0.15 | 0.14 |
| Ramachandran plot | | | | |
| Favored (%) | 95.76 | 99.08 | 99.01 | 99.67 |
| Allowed (%) | 4.24 | 0.92 | 0.99 | 0.33 |
| Disallowed (%) | 0.00 | 0.00 | 0.00 | 0.00 |

**Extended Data Table 2 | Cryo-EM data collection, refinement and validation statistics 2**

| | #5 The cryo-EM structure of the RAD51 filament bound to the nucleosome (EMDB-36444) (PDB 8JNF) | #6 The cryo-EM structure of the RAD51 L1 and L2 loops bound to the linker DNA with the sticky end of the nucleosome (EMDB-38230) (PDB 8XBV) | #7 The cryo-EM structure of the RAD51 N-terminal lobe domain bound to the histone H4 tail of the nucleosome (EMDB-38231) (PDB 8XBW) | #8 The cryo-EM structure of the RAD51 L2 loop bound to the linker DNA with the blunt end of the nucleosome (EMDB-38232) (PDB 8XBX) | #9 The cryo-EM structure of the RAD51 L1 and L2 loops bound to the linker DNA with the blunt end of the nucleosome (EMDB-38233) (PDB 8XBY) |
|---|---|---|---|---|---|
| **Data collection and processing** | | | | | |
| Magnification | 81,000× | 81,000× | 81,000× | 81,000× | 81,000× |
| Voltage (kV) | 300 | 300 | 300 | 300 | 300 |
| Electron exposure (e–/Å$^2$) | 1.51 | 1.50 | 1.50 | 1.50 | 1.50 |
| Defocus range (μm) | -1.0 to -2.5 | -1.0 to -2.5 | -1.0 to -2.5 | -1.0 to -2.5 | -1.0 to -2.5 |
| Pixel size (Å) | 1.06 | 1.06 | 1.06 | 1.06 | 1.06 |
| Symmetry imposed | C1 | C1 | C1 | C1 | C1 |
| Initial particle images (no.) | 522,596 | 4,476,315 | 4,476,315 | 2,044,695 | 4,476,315 |
| Final particle images (no.) | 53,864 | 16,444 | 411,212 | 42,054 | 77,077 |
| Map resolution (Å) | 6.8 | 7.6 | 2.9 | 4.4 | 7.8 |
| FSC threshold | 0.143 | 0.143 | 0.143 | 0.143 | 0.143 |
| Map resolution range (Å) | 4.7 to 13 | 4.2 to 13 | 2.8 to 7.1 | 3.2 to 13 | 4.1 to 17 |
| | | | | | |
| **Refinement** | | | | | |
| Initial model used (PDB code) | PDB 5Y0C, 7OHC, 5NWL, 1WD1, and 5H1C | PDB 7OHC and 5NWL | PDB 5Y0C, 7OHC, and 5NWL | PDB 7OHC and 5NWL | PDB 7OHC and 5NWL |
| Model resolution (Å) | 7.4 | 35 | 5.8 | 8.2 | 32 |
| FSC threshold | 0.5 | 0.5 | 0.5 | 0.5 | 0.5 |
| Map sharpening $B$ factor (Å$^2$) | 1.83 | -13.1 | -41.5 | -7.13 | -1.79 |
| Model composition | | | | | |
| Non-hydrogen atoms | 23975 | 3947 | 2671 | 5873 | 5873 |
| Protein residues | 2446 | 442 | 242 | 663 | 663 |
| Nucleotide | 246 | 27 | 36 | 38 | 38 |
| $B$ factors (Å$^2$) | | | | | |
| Protein | 551.15 | 611.31 | 217.92 | 880.80 | 1706.29 |
| Nucleotide | 445.66 | 497.79 | 156.74 | 426.52 | 1568.24 |
| R.m.s. deviations | | | | | |
| Bond lengths (Å) | 0.007 | 0.005 | 0.006 | 0.006 | 0.006 |
| Bond angles (°) | 1.238 | 1.096 | 1.043 | 1.142 | 1.143 |
| Validation | | | | | |
| MolProbity score | 1.90 | 1.82 | 1.57 | 1.65 | 1.72 |
| Clashscore | 25.64 | 21.08 | 11.47 | 13.70 | 16.34 |
| Poor rotamers (%) | 0.40 | 0.00 | 0.00 | 0.00 | 0.00 |
| Ramachandran plot | | | | | |
| Favored (%) | 99.04 | 99.08 | 99.15 | 99.08 | 99.08 |
| Allowed (%) | 0.96 | 0.92 | 0.85 | 0.92 | 0.92 |
| Disallowed (%) | 0.00 | 0.00 | 0.00 | 0.00 | 0.00 |

**Extended Data Table 3 | *Saccharomyces cerevisiae* strains used in this study**

| Strain | Genotype | Source |
|---|---|---|
| BY4741 | *MATa his3Δ1 leu2Δ0 met15Δ0 ura3Δ0* | ATCC201388 |
| YTS2 | *MATa his3Δ1 leu2Δ0 met15Δ0 ura3Δ0 rad51Δ::kanMX6* | This study |
| YTS6 | *MATa his3Δ1 leu2Δ0 met15Δ0 ura3Δ0 rad51Δ::RAD51-URA3* | This study |
| YTS10 | *MATa his3Δ1 leu2Δ0 met15Δ0 ura3Δ0 rad51Δ::rad51Δ-URA3* | This study |
| YTS14 | *MATa his3Δ1 leu2Δ0 met15Δ0 ura3Δ0 rad51Δ::RAD51K122A-URA3* | This study |
| YTS19 | *MATa his3Δ1 leu2Δ0 met15Δ0 ura3Δ0 rad51Δ::RAD51K128A-URA3* | This study |
| YTS22 | *MATa his3Δ1 leu2Δ0 met15Δ0 ura3Δ0 rad51Δ::RAD51K122A/K128A-URA3* | This study |

# Reporting Summary

## Statistics

For all statistical analyses, confirm that the following items are present in the figure legend, table legend, main text, or Methods section.

| n/a | Confirmed | |
|---|---|---|
| ☐ | ☒ | The exact sample size (*n*) for each experimental group/condition, given as a discrete number and unit of measurement |
| ☐ | ☒ | A statement on whether measurements were taken from distinct samples or whether the same sample was measured repeatedly |
| ☐ | ☒ | The statistical test(s) used AND whether they are one- or two-sided *Only common tests should be described solely by name; describe more complex techniques in the Methods section.* |
| ☐ | ☒ | A description of all covariates tested |
| ☐ | ☒ | A description of any assumptions or corrections, such as tests of normality and adjustment for multiple comparisons |
| ☐ | ☒ | A full description of the statistical parameters including central tendency (e.g. means) or other basic estimates (e.g. regression coefficient) AND variation (e.g. standard deviation) or associated estimates of uncertainty (e.g. confidence intervals) |
| ☐ | ☒ | For null hypothesis testing, the test statistic (e.g. *F*, *t*, *r*) with confidence intervals, effect sizes, degrees of freedom and *P* value noted *Give P values as exact values whenever suitable.* |
| ☒ | ☐ | For Bayesian analysis, information on the choice of priors and Markov chain Monte Carlo settings |
| ☒ | ☐ | For hierarchical and complex designs, identification of the appropriate level for tests and full reporting of outcomes |
| ☒ | ☐ | Estimates of effect sizes (e.g. Cohen's *d*, Pearson's *r*), indicating how they were calculated |

*Our web collection on statistics for biologists contains articles on many of the points above.*

## Software and code

Policy information about availability of computer code

| Data collection | EPU 3.1 |
|---|---|
| Data analysis | Relion 4.0-beta-2, MotionCor2 1.4.0, ctffind-4.1.14, UCSF ChimeraX 1.2, 1.3, 1.4, 1.5 and 1.6, ISOLDE 1.3 and 1.4, Coot 0.9.8.1, PHENIX 1.20, ImageJ 1.53e and 1.53f51, ImageQuant TL ver.8.1, Topaz, R 4.3.2, Python 3.11.7. |

For manuscripts utilizing custom algorithms or software that are central to the research but not yet described in published literature, software must be made available to editors and reviewers. We strongly encourage code deposition in a community repository (e.g. GitHub). See the Nature Portfolio guidelines for submitting code & software for further information.

## Data

Policy information about availability of data

All manuscripts must include a data availability statement. This statement should provide the following information, where applicable:
- Accession codes, unique identifiers, or web links for publicly available datasets
- A description of any restrictions on data availability
- For clinical datasets or third party data, please ensure that the statement adheres to our policy

The cryo-EM structures and the atomic models of the RAD51-nucleosome complexes have been deposited in the Electron Microscopy Data Bank (EMDB) and the Protein Data Bank (PDB), respectively. The accession codes are as follows: EMD-38228 and PDB ID 8XBT for the cryo-EM structure of the octameric RAD51 ring bound to the nucleosome with the linker DNA binding, EMD-36442 and PDB ID 8JND for the cryo-EM structure of the nonameric RAD51 ring bound to the

nucleosome with the linker DNA binding, EMD-38229 and PDB ID 8XBU for the cryo-EM structure of the decameric RAD51 ring bound to the nucleosome with the linker DNA binding, EMD-36443 and PDB ID 8JNE for the cryo-EM structure of the decameric RAD51 ring bound to the nucleosome without the linker DNA binding, EMD-36444 and PDB ID 8JNF for the cryo-EM structure of the RAD51 filament bound to the nucleosome, EMD-38230 and PDB ID 8XBV for the cryo-EM structure of RAD51 L1 and L2 loops bound to the linker DNA with the sticky end of the nucleosome, EMD-38231 and PDB ID 8XBW for the cryo-EM structure of the RAD51 N-terminal lobe domain bound to the histone H4 tail of the nucleosome, EMD-38232 and PDB ID 8XBX for the cryo-EM structure of RAD51 L2 loop bound to the linker DNA with the blunt end of the nucleosome, EMD-38233 and PDB ID 8XBY for the cryo-EM structure of RAD51 L1 and L2 loops bound to the linker DNA with the blunt end of the nucleosome. Uncropped images are shown in Supplementary Figs. 2-5. Source data for the quantification are provided with this paper.

## Research involving human participants, their data, or biological material

Policy information about studies with human participants or human data. See also policy information about sex, gender (identity/presentation), and sexual orientation and race, ethnicity and racism.

| | |
|---|---|
| Reporting on sex and gender | N/A |
| Reporting on race, ethnicity, or other socially relevant groupings | N/A |
| Population characteristics | N/A |
| Recruitment | N/A |
| Ethics oversight | N/A |

Note that full information on the approval of the study protocol must also be provided in the manuscript.

# Field-specific reporting

Please select the one below that is the best fit for your research. If you are not sure, read the appropriate sections before making your selection.

☒ Life sciences          ☐ Behavioural & social sciences          ☐ Ecological, evolutionary & environmental sciences

For a reference copy of the document with all sections, see nature.com/documents/nr-reporting-summary-flat.pdf

# Life sciences study design

All studies must disclose on these points even when the disclosure is negative.

| | |
|---|---|
| Sample size | Sample sizes in all assays were three independent experiments. Sample size calculation was not conducted, as three independent experiments are sufficient for the biological statical analysis. |
| Data exclusions | No data was excluded from the analysis. |
| Replication | Three independent experiments were performed in the electrophoretic mobility shift assays, the western blots and the DNA damage sensitivity assays with S. cerevisiae. All replicated experiments were performed successfully. Cryo-EM analysis was conducted once because the final map already represented the average of a large number of images. |
| Randomization | S. cerevisiae strains were randomly allocated to the DNA damage sensitivity assays. There were no other experiments which required randomization. |
| Blinding | Since there was no subjective allocation in our investigation, blinding was irrelevant to this study |

# Reporting for specific materials, systems and methods

We require information from authors about some types of materials, experimental systems and methods used in many studies. Here, indicate whether each material, system or method listed is relevant to your study. If you are not sure if a list item applies to your research, read the appropriate section before selecting a response.

## Materials & experimental systems

| n/a | Involved in the study |
|---|---|
| ☐ | ☒ Antibodies |
| ☒ | ☐ Eukaryotic cell lines |
| ☒ | ☐ Palaeontology and archaeology |
| ☒ | ☐ Animals and other organisms |
| ☒ | ☐ Clinical data |
| ☒ | ☐ Dual use research of concern |
| ☒ | ☐ Plants |

## Methods

| n/a | Involved in the study |
|---|---|
| ☒ | ☐ ChIP-seq |
| ☒ | ☐ Flow cytometry |
| ☒ | ☐ MRI-based neuroimaging |

## Antibodies

| Antibodies used | Anti-S. cerevisiae Rad51 rabbit antibody (BioAcademia, cat #62-101) , HRP-conjugated anti-rabbit IgG (1:5,000; Merck; NA9340), and HRP-conjugated anti-tubulin alpha (Bio-Rad, cat #MCA77P) were used in the western blots. |
|---|---|
| Validation | Anti-S. cerevisiae Rad51 rabbit antibody (BioAcademia, cat #62-101) Western blot of crude extract of S. cerevisiae was performed.https://www.bioacademia.co.jp/html/upload/save_image/0318174257_623446113b0d3.pdf <br><br> HRP-conjugated anti-tubulin alpha (Bio-Rad, cat #MCA77P) western blot of HeLa whole cell lysate was performed,https://www.bio-rad-antibodies.com/monoclonal/yeast-tubulin-alpha-antibody-yl1-2-mca77.html?f=hrp |

