## [Peer Review File · Nature]

Manuscript Title: Cryo-EM structures of RAD51 assembled on nucleosomes containing a DSB site

Reviewer Comments & Author Rebuttals

Reviewer Reports on the Initial Version:

Referees' comments:

Referee #1:

Shioi et al. report structures of human Rad51 complexed with nucleosome DNA. Previously it was reported that RAD51 can unwrap nucleosomes in an ATP-dependent manner (Dupaigne et al., 2008, PubMed ID: 18982066; Ref. #46). It is gratifying to visualize the structures of RAD51 unwrapping nucleosome and the freed DNA forming RAD51-DNA filaments. The authors also have identified two conserved positively charged residues in the N-terminal domain of Rad51 that are necessary for nucleosome binding and cell survival upon DNA damage. Moreover, mutation of one of these two key residues (K70I) has been found in cancer patients. These results are exciting.

In addition to the ATP-dependent DNA-bound filamentous structure of RAD51, a number of different ring structures of RAD51 were also observed to bind nucleosome linker DNA with or without nucleotide cofactors (ADP, AMPPNP or ATP). However, in the main text, the resolution of the structures resolution is not mentioned at all. According to the Extended Data figures, the RAD51 parts in all reported structures appear to have resolutions no better than 6 Å. Perhaps because of the limited resolution, interactions between RAD51 (filament or ring form) and histone proteins are not specified and not tested by functional assays.

The following structural and functional interpretations would benefit from further explanation.

1. Are the ring-form structures of RAD51 (Fig. 1a-e) functionally relevant? The essential residues (K64 and K70 in human RAD51, and K122 and K128 in yeast RAD51) for nucleosome DNA binding by the RAD51 rings are also involved in nucleosome disassembly by the RAD51 filament (Figs. 1f, 2c). The biochemical analyses of nucleosome binding and cell survival upon DNA damage do not distinguish the function of ring form from the filament form of RAD51. Given the variability of the ring size (octameric, nonameric, and decameric) and the lack of dependence of ADP/ATP in the ring formation, one wonders if such RAD51 rings form without DNA, like the crystal structure of P. furiosus RAD51. Have the authors checked RAD51 structures without nucleosomes?
2. The authors report that the RAD51 ring binds the dsDNA-ssDNA junction region. Are the DNA ends well defined by the cryoEM map volumes? Are all 153 bp of DNA fully observed in the nucleosome-RAD51 ring structures? Have the authors examined RAD51 structures with nucleosomes of 156-bp dsDNA and without the 3-nt ssDNA portion?
3. The title states that "RAD51 assembles on DSB (double-strand break) site in chromatin". But there

is no evidence or reference supporting the statement. Is a DSB necessary for RAD51 binding to nucleosome? The DNA damaging agents (MMS, CPT and HU) used in the cell survival assays (Fig. 3c) do not directly cause DSBs. Instead, MMS causes base modification and replication fork stalling, CPT causes single-strand breaks, and HU causes replication stress. If the conclusion is that nucleosome binding by RAD51 is necessary for DSB repair, why did the authors not use effective and direct means to generate DSBs, such as γ -radiation and dsDNA cleavage agents? According to the existing data, RAD51 and nucleosome binding are required for DNA repair by homologous recombination.

Referee #2:

During double-strand break (DSB) repair, DNA ends are resected leaving 3' overhangs which are bound by RAD51. Multiple copies of RAD51 assemble on DNA to form extended filaments. RAD51 filaments are known to cause nucleosome/chromatin disassembly but structural mechanisms are not clearly established.

Here Shioi et al. report several low resolution cryo-EM structures of RAD51 bound to nucleosomes with one DNA linker with a 3' overhang. These structures were solved in the absence of nucleotide or with ATP, ADP, or the non-hydrolyzable AMP-PNP, the later previously shown to activate RAD51. In several structures, octameric, nonameric, or decameric rings of RAD51 were observed bound to nucleosomal DNA and to the DNA end. In these structures a region of the N-terminal lobe of RAD51 including K64 and K70 is near nucleosomal DNA. These residues are conserved from yeast to man and simultaneous mutation of both these residues leads to loss of nucleosome binding and increased sensitivity to agents that cause DSBs in yeast. An additional structure was observed in which the RAD51 ring bound to the nucleosome without approaching the DNA linker. The authors emphasize a potential interaction with the H4 N-terminal tail in this structure that might be regulated by histone post-translational modifications but have no data to support any role for the tail at this time. A final structure was solved in the presence of AMP-PNP that shows RAD51 filament formation on DNA unwrapped from the nucleosome, presumably an intermediate in nucleosome disassembly. Interestingly the N-terminal lobe approaches histones H2A and H2B where DNA is unwrapped, but again the functional significance of this is unclear as no additional experiments are performed.

The authors propose a mechanism in which RAD51 binds the nucleosome away from the resected DNA end, then the ring repositions to the DNA end, before forming a filament and unwrapping DNA from the histone octamer. This model is compelling and supported by the structures. But ultimately, especially given the resolution of the structures, many elements of the model are highly speculative (i.e., H4 tail binding and regulation, different ring forms and conversion between them or to the filament form, H2A/H2B-RAD51 interaction, nucleosome dimer formation), and the manuscript would be much stronger with more careful functional dissection of RAD51 function on chromatin. Higher confidence in one or more of these novel mechanistic speculations would elevate the impact of the manuscript and make it of broad interest to DNA damage repair and chromatin/epigenetics fields.

Major comments:

1. The authors observe different subsets of complexes without nucleotide or in the presence of ATP, ADP, and AMP-PNP. However, the significance of the specific subsets of structures observed in each nucleotide state are not clearly discussed outside of the AMP-PNP nucleotide claimed to activate RAD51. How is the mechanism supported by the subsets of structures observed in each nucleotide state? Is there a functional difference between 8-, 9- or 10-subunit rings or a difference in how nucleosomes are recognized? Nucleosome binding experiments were performed in Fig. 3b in the presence of AMP-PNP. Are similar results observed (e.g., with K64A and K70A mutations) in the absence of nucleotide or with ATP or ADP in which filaments are not formed? Are nucleosomes disassembled in the AMP-PNP gel shift experiments? What about with ATP or ADP?

2. The authors emphasize the H4 tail in one of their structures (Fig. 2d) because it is near the N-terminal lobe. A paper is cited (ref. 21) that shows that overexpression of SET8 (an H4K20 methyltransferase) decreases RAD51 accumulation. This is claimed as supporting evidence for a direct regulatory role of the H4 tail in RAD51's chromatin activity. However, it is much more likely that this is merely a consequence of 53BP1 recruitment by H4K20me2, which then regulates end resection upstream of RAD51. It is unclear if the EM maps are of sufficient resolution to support a direct H4 tail-RAD51 interaction and no functional experiments are performed to test this hypothesis. Is RAD51's nucleosome binding or disassembly affected by H4 tail deletions, mutations, or even H4K20me2? Is there good charge complementarity on RAD51 to enable binding to the basic H4 tail?

3. Are K64/K70 nucleosome DNA interactions similar in different RAD51 subunits within each structure and between structures? Some further discussion of whether this is one binding mechanism used by every protomer or a flexible binding mechanism in which protomers bind nucleosomal DNA in different ways would add clarity to the manuscript.

4. On lines 165-167, the authors state that the L1 loop may directly interact with the dsDNA-ssDNA junction. Is there sufficient resolution to observe this? If so, can the authors test mutations of the L1 loop in a functional assay on nucleosomes (i.e., binding and/or disassembly with and without end resection)?

5. In the filament structure, a RAD51 protomer contacts the H2A-H2B dimer. Can the authors test whether this is functionally relevant? Can mutations be made in RAD51 or histones that would disrupt this interaction without changing nucleosome integrity or other RAD51 functions? This is a really interesting finding, but it is unclear whether there is any relevance RAD51 activity.

6. RAD51 cancer mutations and phosphosites are introduced in the discussion section. Functional dissection of cancer mutations and/or phosphomimetic mutations on RAD51 nucleosome binding and disassembly would add strength to the manuscript.

Minor comments:

1. In Suppl. Fig. 4b, multiple gel shifted bands are observed with wild-type RAD51, and all but one are essentially lost in the single mutants. What are these extra bands that are lost? Can the bands be

correlated with the different structures in any way (maybe by looking at gel shifts in different nucleotide states)?

2. Consider removing “first” from title.

3. No EM density is shown for any of the regions of the structures discussed in the paper. Are positions of modeled K64/K70 side chains or even main chains clearly defined in the maps? What about the L1 loops? Some more transparency of the model-model correlation is necessary.

4. Can the authors number promoters in Fig. 5a,b? This would help discussion in manuscript that is hard to correlate with the figure.

5. Ref. 20 suggests interaction between RAD51 and DNA involving K64, but this is not observed in this manuscript. Can the authors comment on this discrepancy to previously reported data?

6. Consider citing <https://doi.org/10.1093/nar/gkw920> and <https://doi.org/10.1371/journal.pone.0003643> in addition to ref. 46. A further discussion of these single-molecule FRET studies in light of structures could add strength to the manuscript.

Statistical analysis of gel shift data is well described and appropriate.

Referee #3:

This manuscript reports on several cryo-EM structures of the human RAD51 recombinase in complex with an NCP. Obtained structures, and biochemical and yeast-based cell survival assays shed light on the molecular mechanisms by which a RAD51 filament may be initiated for assembly in the context of chromatin. Overall, this manuscript addresses long-standing knowledge gaps and provides important new information on the mechanisms of RAD51 protomer “storage” and filament assembly in cells. This study likely will be of high impact and is predicted to move the field forward.

The investigators show that RAD51 protomers form multiple complexes with a nucleosome, including several rings and a filament on extended ssDNA. They also identify the RAD51 N-terminal Lobe Domain (NLD) as the nucleosome-binding module, and, presumably, as in direct contact with nucleosomal DNA. Conserved residues in the NLD (Lys64 and Lys70), which in the rings and in the filament are located near the nucleosomal DNA, are critical for NCP-binding in EMSAs (human RAD51) and for cell survival (yeast RAD51).

Based on the obtained structures, the investigators speculate that the RAD51 ring without the bound linker DNA may be the primary nucleosome binding form and, as such, represent the initial RAD51 assembly on chromatin (possibly through contact with the H4 N-terminal tail). RAD51 rings with linker DNA incorporated through their central hole show evidence of dsDNA/ssDNA junctions located close to RAD51’s L1 loop, in support of its previously identified DNA binding activity. Through this configuration, the investigators speculate that detected dsDNA/ssDNA junctions may prime filament formation.

More specific comments:

Fig. 1a: C-terminal residue(s) of the NLDs should be indicated; L1 loop residues should also be indicated.

Fig. 3c, + delta rad51: It is unclear why a "+" is shown here.

Line 130: It is stated that the RAD51K70A mutant exhibits a slight but clear defect in NCP-binding. The statistics are not shown but should be added to Fig. 3. Similarly, is the diminished DNA binding activity by the double mutant statistically significant?

The L2 region is not discussed. Is this region not discernible in any of the structures? L2 residues are apparent in some of the previously published structures. One would expect that this region also shares some contact points with the DNA peeled off the nucleosome.

Lys64 and Lys70 are located within and close to the previously identified HhH motif, and also close to the polymerization motif. There is some concern that, in addition to NCP-binding, other RAD51 attributes (i.e., polymerization) may lead to the phenotype observed in yeast (note that even for the double mutant NCP-binding is not fully abrogated). Has this been tested?

Have the investigators anticipated to obtain structures with the RAD51 single-site mutants that appear fully and almost fully capable of shifting NCPs in EMSAs?

Author Rebuttals to Initial Comments:

Referee #1:

General comment)

Shioi et al. report structures of human Rad51 complexed with nucleosome DNA. Previously it was reported that RAD51 can unwrap nucleosomes in an ATP-dependent manner (Dupaigne et al., 2008, PubMed ID: 18982066; Ref. #46). It is gratifying to visualize the structures of RAD51 unwrapping nucleosome and the freed DNA forming RAD51-DNA filaments. The authors also have identified two conserved positively charged residues in the N-terminal domain of Rad51 that are necessary for nucleosome binding and cell survival upon DNA damage. Moreover, mutation of one of these two key residues (K70I) has been found in cancer patients. These results are exciting.

In addition to the ATP-dependent DNA-bound filamentous structure of RAD51, a number of different ring structures of RAD51 were also observed to bind nucleosome linker DNA with or without nucleotide cofactors (ADP, AMPPNP or ATP). However, in the main text, the resolution of the structures resolution is not mentioned at all. According to the Extended Data figures, the RAD51 parts in all reported structures appear to have resolutions no better than 6 Å. Perhaps because of the limited resolution, interactions between RAD51 (filament or ring form) and histone proteins are not specified and not tested by functional assays.

The following structural and functional interpretations would benefit from further explanation.

Reply)

Thank you very much for these constructive comments. We revised the manuscript according to this reviewer's suggestions.

Comment 1)

Are the ring-form structures of RAD51 (Fig. 1a-e) functionally relevant? The essential residues (K64 and K70 in human RAD51, and K122 and K128 in yeast RAD51) for nucleosome DNA binding by the RAD51 rings are also involved in nucleosome disassembly by the RAD51 filament (Figs. 1f, 2c). The biochemical analyses of nucleosome binding and cell survival upon DNA damage do not distinguish the function of ring form from the filament form of RAD51. Given the variability of the ring size (octameric, nonameric, and decameric) and the lack of dependence of ADP/ATP in the ring formation, one wonders if such RAD51 rings form without DNA,

like the crystal structure of *P. furiosus* RAD51. Have the authors checked RAD51 structures without nucleosomes?

Reply)

As this reviewer suggested, we obtained the cryo-EM micrograph images of wild type RAD51 and the RAD51K64A•K70A mutant without nucleosomes. We then found that both RAD51 and RAD51K64A•K70A efficiently form helical filament structures, but not the ring forms, in the absence of the nucleosome. These results are consistent with the idea that the formation of RAD51 rings may be responsible for its nucleosome binding. This new finding is presented in the new Extended Data Fig. 1d and described in the revised text (p.6, l.32-p.7, l.1).

In addition, we performed the nucleosome binding assay in the absence and presence of ADP, ATP, and AMP-PNP, and assigned the bands corresponding to the RAD51 ring-nucleosome complexes with the bound linker DNA, two RAD51 rings bound to the nucleosome, the RAD51 filament bound to the nucleosome, and the RAD51 filament without the nucleosome, based on the cryo-EM observations. These assignments allowed us to assess the nucleosome binding profiles of the RAD51 mutants, and distinguish the functions of the ring and filament forms of RAD51. These results are presented in the new Extended Data Fig. 6a and described in the new Results sections “Cryo-EM structures of the RAD51 ring and filament forms in complex with nucleosomes”, “The N-terminal lobe domain of RAD51 functions as the nucleosome-binding module”, “The N-terminal lobe domain of RAD51 interacts with the N-terminal tail of histone H4 in a decameric RAD51 ring without the bound linker DNA”, and “Mutational analyses of the RAD51 NLD-nucleosome interaction”.

Comment 2)

The authors report that the RAD51 ring binds the dsDNA-ssDNA junction region. Are the DNA ends well defined by the cryoEM map volumes? Are all 153 bp of DNA fully observed in the nucleosome-RAD51 ring structures? Have the authors examined RAD51 structures with nucleosomes of 156-bp dsDNA and without the 3-nt ssDNA portion?

Reply)

We performed the focused refinement around the DNA end of the nucleosome complexed with the RAD51 nonameric ring. This allowed the visualization of the cryo-EM map volumes of the nucleosomal DNA end. In addition, the cryo-EM map volumes of the RAD51 L1 loop, which is the catalytic center for homologous pairing, are now clear. Another catalytic center, the RAD51 L2 loop, is also partially

visualized, although the L1 loop seems to be the major ssDNA binding site. These results explain how the RAD51 L1 and L2 loops bind to the ssDNA region in the central hole of the RAD51 ring. Including these new data, we replaced most of the structural figures, and the new results obtained by the detailed analyses of the structures are discussed in the new Results section “The RAD51 ring captures the linker DNA containing the DSB near the nucleosome”.

According to this reviewer’s suggestion, we reconstituted the nucleosome containing one linker DNA with the thirteen base-pair dsDNA without the ssDNA overhang. We then determined the cryo-EM structure of RAD51 complexed with the nucleosome containing this dsDNA linker. Surprisingly, in this new structure, we found that the RAD51 L2 loop, but not the L1 loop, may be the major DNA binding loop for dsDNA. These new results may explain the functional differences between the L1 and L2 loops in the RAD51 ring: the L2 loop preferentially contacts the dsDNA region of the linker DNA, and the L1 loop prefers to bind the ssDNA region produced as a consequence of the DNA resection. These novel structures are presented in the new Fig. 5, and this hypothesis is discussed in the new Results section “The RAD51 ring captures the linker DNA containing the DSB near the nucleosome”.

Comment 3)

The title states that “RAD51 assembles on DSB (double-strand break) site in chromatin”. But there is no evidence or reference supporting the statement. Is a DSB necessary for RAD51 binding to nucleosome? The DNA damaging agents (MMS, CPT and HU) used in the cell survival assays (Fig. 3c) do not directly cause DSBs. Instead, MMS causes base modification and replication fork stalling, CPT causes single-strand breaks, and HU causes replication stress. If the conclusion is that nucleosome binding by RAD51 is necessary for DSB repair, why did the authors not use effective and direct means to generate DSBs, such as γ -radiation and dsDNA cleavage agents? According to the existing data, RAD51 and nucleosome binding are required for DNA repair by homologous recombination.

Reply)

Thank you very much for this insightful comment. According to this reviewer’s suggestion, we performed the *in vivo* DNA repair assay under conditions with X-ray irradiation. Yeasts are extremely resistant to X-rays, as compared to mammalian cells (up to several hundred-fold). Despite this X-ray-resistant characteristic, in *Saccharomyces cerevisiae* cells, the Rad51 mutant, which is specifically defective in nucleosome binding, is clearly defective in the repair of DSBs induced by X-ray irradiation. In the revised manuscript, we added these new data in the new Fig. 4f and described them in the Results section “Mutations in the RAD51 NLD cause defects in DNA repair in cells”.

Referee #2:

Comment)

During double-strand break (DSB) repair, DNA ends are resected leaving 3' overhangs which are bound by RAD51. Multiple copies of RAD51 assemble on DNA to form extended filaments. RAD51 filaments are known to cause nucleosome/chromatin disassembly but structural mechanisms are not clearly established.

Here Shioi et al. report several low resolution cryo-EM structures of RAD51 bound to nucleosomes with one DNA linker with a 3' overhang. These structures were solved in the absence of nucleotide or with ATP, ADP, or the non-hydrolyzable AMP-PNP, the later previously shown to activate RAD51. In several structures, octameric, nonameric, or decameric rings of RAD51 were observed bound to nucleosomal DNA and to the DNA end. In these structures a region of the N-terminal lobe of RAD51 including K64 and K70 is near nucleosomal DNA. These residues are conserved from yeast to man and simultaneous mutation of both these residues leads to loss of nucleosome binding and increased sensitivity to agents that cause DSBs in yeast. An additional structure was observed in which the RAD51 ring bound to the nucleosome without approaching the DNA linker. The authors emphasize a potential interaction with the H4 N-terminal tail in this structure that might be regulated by histone post-translational modifications but have no data to support any role for the tail at this time. A final structure was solved in the presence of AMP-PNP that shows RAD51 filament formation on DNA unwrapped from the nucleosome, presumably an intermediate in nucleosome disassembly. Interestingly the N-terminal lobe approaches histones H2A and H2B where DNA is unwrapped, but again the functional significance of this is unclear as no additional experiments are performed.

The authors propose a mechanism in which RAD51 binds the nucleosome away from the resected DNA end, then the ring repositions to the DNA end, before forming a filament and unwrapping DNA from the histone octamer. This model is compelling and supported by the structures. But ultimately, especially given the resolution of the structures, many elements of the model are highly speculative (i.e., H4 tail binding and regulation, different ring forms and conversion between them or to the filament form, H2A/H2B-RAD51 interaction, nucleosome dimer formation), and the manuscript would be much stronger with more careful functional dissection of RAD51 function on chromatin. Higher confidence in one or more of these novel mechanistic speculations would elevate the impact of the manuscript and make it of broad interest to DNA damage repair and chromatin/epigenetics fields.

Reply)

Thank you very much for these constructive comments. In accordance with this reviewer's suggestions, we have revised our manuscript as follows.

Major comments:

Comment 1)

The authors observe different subsets of complexes without nucleotide or in the presence of ATP, ADP, and AMP-PNP. However, the significance of the specific subsets of structures observed in each nucleotide state are not clearly discussed outside of the AMP-PNP nucleotide claimed to activate RAD51.

Reply)

We revised the manuscript according to this reviewer's suggestions, as listed below.

Comment 1-1)

•How is the mechanism supported by the subsets of structures observed in each nucleotide state?

Reply)

The decameric RAD51 ring without the bound linker DNA was observed in the presence of ADP, but not AMP-PNP. This may happen because AMP-PNP binding converts this inactive RAD51 ring to an active ring and/or filament. Therefore, the decameric RAD51 ring without the bound linker DNA may be a standby form, which may constantly exist and survey the DNA damages in cells by associating with chromatin. Conversely, the active ring forms bound with the linker DNA are mainly observed in the presence of ATP or AMP-PNP, suggesting that these may be the forms that function in the DSB recognition process. These points are now described in the second paragraph of the Results section "Cryo-EM structures of the RAD51 ring and filament forms in complex with nucleosomes".

Comment 1-2)

•Is there a functional difference between 8-, 9- or 10-subunit rings or a difference in how nucleosomes are recognized?

Reply)

To explore this reviewer's concern, we further analyzed the cryo-EM structures of the RAD51 8-, 9-, and 10-mer rings, and built their models. These models revealed the differences and similarities in the nucleosome binding modes among the active RAD51 8-, 9-, and 10-mer rings. We re-numbered the RAD51 protomers in each ring, with the first RAD51 protomer contacting the nucleosomal DNA proximal to the linker DNA designated as No.1, and numbered the protomers in a clockwise manner. We found that the RAD51 NLD of the No.1 protomer (RAD51 No.1 NLD) consistently binds in proximity to the nucleosomal DNA entry/exit regions at the 137th base-pair position for the octameric ring, 138th for the nonameric ring, and 128th for the decameric ring from the distal end of the nucleosomal DNA. These findings suggest that the RAD51 No.1 NLD may preferentially bind near the nucleosomal entry/exit DNA region across all three ring configurations, with a similar nucleosomal DNA binding mode. The Lys64 and Lys70 residues of the NLDs appear to play a pivotal role in the nucleosomal DNA binding, while additional RAD51 NLDs engage the nucleosomal DNA at various positions. The No.6 NLD in the octameric ring, No.7 NLD in the nonameric ring, and No.3 NLD in the decameric ring bind at the 72nd, 77th, and 55th base-pair positions from the distal end of the DNA, respectively. In addition, in the decameric RAD51 ring with the linker DNA binding, the structures of the No.1 and No.3 protomers are different. In the context of the RAD51 decameric ring bound with the linker DNA, the L1 loop region of the No.1 protomer also contributes to the nucleosomal DNA binding, in concert with the Lys64 and Lys70 residues of the NLD. The sequential contacts at the 136th, 134th, and 132nd base-pair positions from the distal end of the nucleosomal DNA by the L1 loops of the RAD51 No.1, No.2, and No.3 protomers, respectively, are distinctive to the decameric ring. These interactions were not observed in the octameric and nonameric rings, suggesting that the RAD51 decameric ring with the linker DNA binding may be an intermediate form in the structural transition from the ring to the filament. These new findings are presented in the new Fig. 2b-f, and described in the Results section "The N-terminal lobe domain of RAD51 functions as the nucleosome-binding module".

Comment 1-3)

•Nucleosome binding experiments were performed in Fig. 3b in the presence of AMP-PNP. Are similar results observed (e.g., with K64A and K70A mutations) in the absence of nucleotide or with ATP or ADP in which filaments are not formed?

Reply)

As this reviewer suggested, we performed the nucleosome binding experiments with AMP-PNP, ATP, and ADP, and successfully assigned the bands corresponding to one RAD51 ring complexed with the nucleosome with the linker DNA binding, two RAD51 rings complexed with the nucleosome with and without linker DNA binding, and the RAD51 filament complexed with the nucleosome, by comparing the observed cryo-EM structures under each set of nucleotide conditions. We also detected the RAD51 filament formed on naked DNA. These data are presented in the new Extended Data Fig. 6a. We then tested the nucleosome binding activities of the RAD51 K64A, K70A, and K64A•K70A mutants under conditions with ADP, which allow RAD51 ring but not filament formation. We surprisingly found that the nucleosome binding activity of RAD51 is drastically reduced in the RAD51 K64A, K70A, and K64A•K70A mutants, although it is partially retained in the presence of AMP-PNP, which mainly facilitates RAD51 filament formation. These results strongly suggest that the RAD51 Lys64 and Lys70 residues play important roles in the nucleosome binding by the RAD51 rings. These new data are presented in the new Fig. 4c, d, and the details are discussed in the new Results section “Mutational analyses of the RAD51 NLD-nucleosome interaction”.

Comment 1-4)

•Are nucleosomes disassembled in the AMP-PNP gel shift experiments? What about with ATP or ADP?

Reply)

In the AMP-PNP gel shift experiments, we detected bands corresponding to RAD51 filament formation on the partially disassembled nucleosome and the disassembled naked DNA. These bands corresponding to the RAD51 filaments may not be observed in the presence of ADP. The nucleotide-dependent formation of the RAD51-nucleosome complexes that are separately detected by EMSA are presented in Extended Data Fig. 6a, and described at the end of the Results section “Cryo-EM structures of the RAD51 ring and filament forms in complex with nucleosomes”. In addition, we prepared the RAD51E59R mutant. In the cryo-EM structure of the RAD51 filament-nucleosome complex, the RAD51 Glu59 residue may directly contact the exposed histone H2A-H2B surface by partial nucleosomal DNA peeling by RAD51. We found that the RAD51E59R mutant specifically decreases the RAD51 filament-nucleosome complex formation in our EMSA. These new data are presented in the new Fig. 6, and the results are described in the Results section “The RAD51 filament formed in the nucleosome”.

Comment 2)

The authors emphasize the H4 tail in one of their structures (Fig. 2d) because it is near the N-terminal lobe. A paper is cited (ref. 21) that shows that overexpression of SET8 (an H4K20 methyltransferase) decreases RAD51 accumulation. This is claimed as supporting evidence for a direct regulatory role of the H4 tail in RAD51's chromatin activity. However, it is much more likely that this is merely a consequence of 53BP1 recruitment by H4K20me2, which then regulates end resection upstream of RAD51. It is unclear if the EM maps are of sufficient resolution to support a direct H4 tail-RAD51 interaction and no functional experiments are performed to test this hypothesis. Is RAD51's nucleosome binding or disassembly affected by H4 tail deletions, mutations, or even H4K20me2? Is there good charge complementarity on RAD51 to enable binding to the basic H4 tail?

Reply)

Thank you very much for this insightful comment. As this reviewer pointed out, overexpression of SET8 may simply decrease RAD51 accumulation through a consequence of 53BP1 recruitment. In the revised manuscript, we removed the corresponding sentences. In our revised experiments, we performed focused refinement around the H4 N-terminal tail and successfully improved the cryo-EM map of the H4 N-terminal tail extending to the acidic region of the RAD51 NLD. This suggests that the NLD has good charge complementarity for binding the basic H4 N-terminal tail. According to this reviewer's suggestion, we tested the RAD51-nucleosome binding with the nucleosome lacking the N-terminal 19 residues of H4, and found that the H4 N-terminal tail is actually important for the RAD51-nucleosome binding. As a control, we confirmed that the nucleosome lacking the N-terminal 15 residues of H4 did not affect the RAD51 binding. These results are presented in the new Fig. 3b, c, and are discussed in the new Results section "The N-terminal lobe domain of RAD51 interacts with the N-terminal tail of histone H4 in a decameric RAD51 ring without the bound linker DNA".

Comment 3)

Are K64/K70 nucleosome DNA interactions similar in different RAD51 subunits within each structure and between structures? Some further discussion of whether this is one binding mechanism used by every protomer or a flexible binding mechanism in which protomers bind nucleosomal DNA in different ways would add clarity to the manuscript.

Reply)

Thank you very much for this important comment. In the revised manuscript, we compared the DNA binding modes of each NLD bound to the nucleosomal DNA in

the octameric, nonameric, and decameric RAD51 rings. As explained in the response to comment 1-2 above, we then found that the nucleosomal DNA binding mode of the RAD51 NLD No.1 protomers is common among the three rings. The other nucleosome binding NLDs in the octameric and nonameric rings also bind to the nucleosomal DNA with a conserved mechanism. Interestingly, we found that the RAD51 NLD No.3 protomer of the decameric RAD51 ring binds differently to the nucleosomal DNA. In addition, the structures of the No.1 and No.3 protomers of the decameric RAD51 ring are drastically different. Conversely, in all cases, the RAD51 Lys64 and Lys70 residues are located near the DNA backbone, and may directly contact the nucleosomal DNA. These new data are presented in the new Fig. 2, and are discussed in the Results section “The N-terminal lobe domain of RAD51 functions as the nucleosome-binding module”.

Comment 4)

On lines 165-167, the authors state that the L1 loop may directly interact with the dsDNA-ssDNA junction. Is there sufficient resolution to observe this? If so, can the authors test mutations of the L1 loop in a functional assay on nucleosomes (i.e., binding and/or disassembly with and without end resection)?

Reply)

In the revised manuscript, we performed the focused refinement around the DNA end of the nucleosome complexed with RAD51. This allowed the cryo-EM map volumes of the nucleosomal DNA end to become visible. In addition, the cryo-EM map volumes of the RAD51 L1 loop are now clear. Another catalytic center in the RAD51 L2 loop was also partially visualized. These new data are shown in the new Fig. 5, and are discussed in the new Results section “The RAD51 ring captures the linker DNA containing the DSB near the nucleosome”.

According to this reviewer’s suggestion, using the structural information, we focused on the L1 loop R235 residue, which is near the linker DNA backbone in the central hole of the nonameric RAD51 ring. We then prepared the RAD51R235D mutant, in which the Arg235 residue is replaced by Asp. The RAD51R235D mutant proficiently forms the helical filament in the presence of AMP-PNP without the DNA and nucleosome. However, in the presence of AMP-PNP, the RAD51R235D mutant is completely defective in nucleosome binding. Consistent with the idea that the L1 loop is the central DNA binding site of RAD51, the RAD51R235D mutant is also defective in DNA binding. These results suggest that the DNA binding by the RAD51 Arg235 residue may play an important role in the nucleosome binding, together with the RAD51 NLD. These new data are presented in the new Extended Data Figs. 1d, 6d, e, and are discussed in the second paragraph of the new Results section “The RAD51 ring captures the linker DNA containing the DSB near the nucleosome”.

Comment 5)

In the filament structure, a RAD51 protomer contacts the H2A-H2B dimer. Can the authors test whether this is functionally relevant? Can mutations be made in RAD51 or histones that would disrupt this interaction without changing nucleosome integrity or other RAD51 functions? This is a really interesting finding, but it is unclear whether there is any relevance RAD51 activity.

Reply)

As this reviewer suggested, we purified the RAD51 E59R mutant, in which the possible H2A-H2B interacting residue, RAD51 Glu59, was replaced by Arg. We then tested the nucleosome binding activity of RAD51 in the presence of AMP-PNP, which allows RAD51 filament formation with nucleosomal DNA peeling. Interestingly, we found that the RAD51 E59R mutant exhibited a specific defect in the RAD51 filament-nucleosome complex formation, without affecting the RAD51 ring-nucleosome binding. These results strongly support the proposal that RAD51 binds the nucleosomal H2A-H2B with the interface containing the Glu59 residue during nucleosome disassembly by filament formation. These new data are presented in the new Fig. 6c, and are discussed in the first paragraph of the Results section "The RAD51 filament formed in the nucleosome".

Comment 6)

RAD51 cancer mutations and phosphosites are introduced in the discussion section. Functional dissection of cancer mutations and/or phosphomimetic mutations on RAD51 nucleosome binding and disassembly would add strength to the manuscript.

Reply)

Thank you very much for this suggestion. In the revised manuscript, we described the cancer-related and phosphosite mutations in the new Results section "Mutational analyses of the RAD51 NLD-nucleosome interaction". To do so, we prepared the RAD51 S67E mutant, in which the phosphosite of the RAD51 Ser67 residue is replaced by the phosphomimetic Glu residue. We then found that the RAD51 S67E mutant is defective in nucleosome binding in the presence of either AMP-PNP or ADP. These novel data are presented in the new Extended Data Fig. 6c, and discussed in the Results section "Mutational analyses of the RAD51 NLD-nucleosome interaction".

Minor comments:

Minor comment 1)

In Suppl. Fig. 4b, multiple gel shifted bands are observed with wild-type RAD51, and all but one are essentially lost in the single mutants. What are these extra bands that are lost? Can the bands be correlated with the different structures in any way (maybe by looking at gel shifts in different nucleotide states)?

Reply)

Thank you very much for this insightful comment. In the revised experiments, we assigned the gel shift bands corresponding to the RAD51 filament-nucleosome complex, the active RAD51 ring-nucleosome complexes, and the active and inactive RAD51 ring-nucleosome complexes (new Extended Data Fig. 6a). We then figured out that the band missing in the RAD51 K64A, K70A, and K64A•K70A mutants was the active RAD51 ring-nucleosome complexes. This suggests that the RAD51 K64A, K70A, and K64A•K70A mutants are defective in nucleosome binding as an active ring form. These results are discussed in the new Results section “Cryo-EM structures of the RAD51 ring and filament forms in complex with nucleosomes” and “Mutational analyses of the RAD51 NLD-nucleosome interaction”.

Minor comment 2)

Consider removing “first” from title.

Reply)

We corrected the title accordingly.

Minor comment 3)

No EM density is shown for any of the regions of the structures discussed in the paper. Are positions of modeled K64/K70 side chains or even main chains clearly defined in the maps? What about the L1 loops? Some more transparency of the model-model correlation is necessary.

Reply)

Thank you very much for this suggestion. In the revised manuscript, we presented the cryo-EM map together with the model structures. In addition, we eliminated the invisible side chains in the cryo-EM map from the model structures. These results are presented in the new Figs. 2, 5, 6.

Minor comment 4)

Can the authors number promoters in Fig. 5a,b? This would help discussion in manuscript that is hard to correlate with the figure.

Reply)

We corrected it accordingly.

Minor comment 5)

Ref. 20 suggests interaction between RAD51 and DNA involving K64, but this is not observed in this manuscript. Can the authors comment on this discrepancy to previously reported data?

Reply)

Thank you very much. In the revised manuscript, we performed the DNA binding assay in the presence of ADP, but not AMP-PNP, and found that the RAD51 Lys64 mutation drastically affected its DNA binding activity. This is consistent with the previous experiments in the presence of ATP, which may be hydrolyzed to ADP. These novel data are presented in the new Fig. 4c, d, and are described in the second paragraph of the new Results section "Mutational analyses of the RAD51 NLD-nucleosome interaction".

Minor comment 6)

Consider

citing <https://doi.org/10.1093/nar/gkw920> and <https://doi.org/10.1371/journal.pone.0003643> in addition to ref. 46. A further discussion of these single-molecule FRET studies in light of structures could add strength to the manuscript.

Reply)

We cited the papers suggested by this reviewer.

Referee #3 :

This manuscript reports on several cryo-EM structures of the human RAD51 recombinase in complex with an NCP. Obtained structures, and biochemical and yeast-based cell survival assays shed light on the molecular mechanisms by which a RAD51 filament may be initiated for assembly in the context of chromatin. Overall, this manuscript addresses long-standing knowledge gaps and provides important new information on the mechanisms of RAD51 protomer “storage” and filament assembly in cells. This study likely will be of high impact and is predicted to move the field forward.

The investigators show that RAD51 protomers form multiple complexes with a nucleosome, including several rings and a filament on extended ssDNA. They also identify the RAD51 N-terminal Lobe Domain (NLD) as the nucleosome-binding module, and, presumably, as in direct contact with nucleosomal DNA. Conserved residues in the NLD (Lys64 and Lys70), which in the rings and in the filament are located near the nucleosomal DNA, are critical for NCP-binding in EMSAs (human RAD51) and for cell survival (yeast RAD51).

Based on the obtained structures, the investigators speculate that the RAD51 ring without the bound linker DNA may be the primary nucleosome binding form and, as such, represent the initial RAD51 assembly on chromatin (possibly through contact with the H4 N-terminal tail). RAD51 rings with linker DNA incorporated through their central hole show evidence of dsDNA/ssDNA junctions located close to RAD51’s L1 loop, in support of its previously identified DNA binding activity. Through this configuration, the investigators speculate that detected dsDNA/ssDNA junctions may prime filament formation.

Reply)

Thank you very much for these comments. According to this reviewer’s comments, we revised the manuscript as described below.

More specific comments:

Comment 1)

Fig. 1a: C-terminal residue(s) of the NLDs should be indicated; L1 loop residues should also be indicated.

Reply)

We added these designations in Fig. 2a.

Comment 2)

Fig. 3c, + delta rad51: It is unclear why a "+" is shown here.

Reply)

We corrected it to empty vector.

Comment 3)

Line 130: It is stated that the RAD51K70A mutant exhibits a slight but clear defect in NCP-binding. The statistics are not shown but should be added to Fig. 3. Similarly, is the diminished DNA binding activity by the double mutant statistically significant?

Reply)

To clarify the mutant characterization, we performed the nucleosome binding assay in the presence of ADP, which allows the formation of the RAD51 ring but not the filament with the nucleosome. We found that the single RAD51K64A and RAD51K70A mutants are both substantially defective in the nucleosome binding, although these defects may not be obvious in the presence of AMP-PNP, which mainly forms the RAD51 filament with the nucleosome. These new experiments shed light on the characteristics of the RAD51 mutants. In addition, we found that the RAD51K70A mutant is defective in nucleosome binding, without affecting the naked DNA binding in the presence of ADP. These novel data are presented in the new Fig. 4c, d, and are discussed in the new Results section "Mutational analyses of the RAD51 NLD-nucleosome interaction".

Comment 4)

The L2 region is not discussed. Is this region not discernible in any of the structures? L2 residues are apparent in some of the previously published structures. One would expect that this region also shares some contact points with the DNA peeled off the nucleosome.

Reply)

In the revised experiments, we performed the focused refinement around the L1 and L2 regions of the RAD51 ring, which binds to the nucleosome and linker DNA. We

then successfully improved the cryo-EM map around the L1 and L2 loops, and the L2 loop region is now partially visualized. In the new map, the L2 loop appears to be located near the DSB end of the linker DNA, together with the L1 loop. In addition, we determined a new structure of the RAD51 ring-nucleosome complex with a dsDNA linker in the presence of ATP. Interestingly, in this complex, the linker DNA binding by the L2 loop is mainly observed near the DSB end. In contrast, the linker DNA binding by the L1 loop may be additional, because structures missing the linker DNA binding by the L1 loop were also obtained. These results suggest the functional differences between the L1 and L2 loops in the RAD51 ring: the L2 loop preferentially contacts the dsDNA region of the linker DNA, and the L1 loop prefers to bind to the ssDNA region produced as a consequence of the DNA resection. These novel results are presented in the new Fig. 5a-c, and extensively described in the Results section “The RAD51 ring captures the linker DNA containing the ssDNA region near the nucleosome”.

Comment 5)

Lys64 and Lys70 are located within and close to the previously identified HhH motif, and also close to the polymerization motif. There is some concern that, in addition to NCP-binding, other RAD51 attributes (i.e., polymerization) may lead to the phenotype observed in yeast (note that even for the double mutant NCP-binding is not fully abrogated). Has this been tested?

Reply)

Thank you very much for this comment. To test the polymer formation activity of the RAD51 K64A•K70A mutant, we performed the cryo-EM analysis. We then found that the RAD51 K64A•K70A mutant forms the helical filament in the presence of AMP-PNP, as efficiently as the wild-type RAD51. This new evidence supports our conclusion that the RAD51 Lys64 and Lys70 residues may have specific functions in nucleosome binding. These novel data are presented in the new Extended Data Fig. 1d, and are described in the text (p.6, l.32-p.7, l.1).

Comment 6)

Have the investigators anticipated to obtain structures with the RAD51 single-site mutants that appear fully and almost fully capable of shifting NCPs in EMSAs?

Reply)

As this reviewer suggested, we prepared the new RAD51 NLD mutant, RAD51 R27A, in which the Arg27 residue is replaced by Ala. The RAD51 Arg27 residue does not face the nucleosomal DNA, but is exposed to the solvent. We then found that the RAD51 R27A mutant is proficient in nucleosome binding in the presence of AMP-PNP. Therefore, the RAD51 R27A mutant serves as a positive control mutant for the nucleosome binding mutants in the RAD51 NLD. These new data are presented in the new Extended Data Fig. 6b, and are described in the text (p.7, l.8-10).

Reviewer Reports on the First Revision:

Referees' comments:

Referee #1:

The authors have addressed my concerns with additional experiments and revision of the text. The title and summary of the paper are well supported by the structures, mutagenesis, in vitro binding and yeast-based repair assays. The main take-home message that the N-terminal lobe domain (NLD) of RAD51 binds to nucleosome and helps to disassemble nucleosomes for DSB repair is clear and convincing.

With the structural models and maps provided by the authors, the following points may be considered to improve the accuracy and focus of the manuscript.

1. As the nucleosome-bound octameric, nonameric, and decameric RAD51 rings share not a single conserved protein-protein or protein-DNA interface when the nucleosome or, a single subunit or NLD of RAD51 is superimposed among them. Fig. 2b-e appear unclear and yet redundant as no cryoEM map defines the K64 and K70 sidechains, and NLD containing these two conserved residues interacts differently with nucleosomes in each panel. The authors may simplify the figure and discussion by consolidating the structural findings and explain that K64 and K70 are chosen for mutagenic studies because they are conserved and have proximity to nucleosome DNA. Why is the conserved K73 not studied in the nucleosome binding and DSB repair?

2. The structures of the Rad51-nucleosome complexes are of medium to low resolution. Even when overall structures appear 3.7–4.7 Å, which is largely due to the well-ordered nucleosome part, Rad51 and Rad51-nucleosome interface are often only at 5-6 Å. In addition to the limited resolutions of structures, these structures were obtained at a low and non-physiological salt concentration (30 mM NaCl), as were all binding studies. Structural interpretations need to take these facts into account. For example, the reported interaction between the histone H4 tail and NLD (Fig. 3) is not as convincing. In the EMSA assay, such complexes are only a small fraction of the “canonical” Rad51-nucleosome complexes independent of the H4 tail. The NLD in the structural model of H4 tail-NLD complex has very weak map volume, and its position is poorly defined and exhibits no contact with the histones. The H4 tail (K16 to and R19) does not contact NLD in the structural models, and instead they bind the nucleosomal DNA well.

3. As the structural data were obtained at 30 mM NaCl, dimerization of the RAD51 filament-nucleosome complexes shown in Fig. 6e is of concern. It does not support the main point of the manuscript, nor is it clear whether the dimerization is an artifact due to the low salt condition or functionally meaningful.

Other points for potential improvement:

1. Each figure can use more labels. For example, panels in Figs. 1 and 2 can use labels, such as octameric, nonameric, or decameric RAD51, and +ADP, +AMPPNP, +ATP or no nucleotide cofactor,

etc. In Fig. 4, adding labels of “+AMPPNP”, “+ADP” to panels a,b and c,d, respectively, will help.

2. Please label complex names in all cryoEM work-flow ED figures. Each figure contains several different complexes. It is not easy to figure out which is what.

3. Is the map resolution 2.89 Å (ED. Fig. 3 and Table 1) a typo or overestimated for the structure of histone H4 tail-NLD complex (complex #7)? The model resolution is reported to be 4.79Å (ED Table 1), nearly 2 Å worse than the map resolution. The best part of the map (histones) appears no better than 3.5 Å, and the NLD is below 5 Å.

4. The labels of the y-axis in Fig. 3b,c contain a typo, “conformation” missing “t”. The quantification shown in the plots appears to quantify the complex of two RAD51 rings bound to one nucleosome and not the 1:1 complex, which is 100% and dominant. The legend and the plot label should clarify this.

5. It is odd that the map resolutions of #5 (RAD51 filament bound to the nucleosome) and #9 complex are 6.77 and 7.81 Å, respectively, and the resulting structural models have the resolution of 4.90 and 4.79 Å. How could the models be so much better than the experimental data (maps)?

6. Resolutions of cryoEM maps and structures do not need double decimals as they are not so precisely estimated.

Referee #2:

The revisions added substantially to the clarity of the manuscript and the functional relevance of the structures. All of my comments and queries were addressed. The study overall is incredibly exciting, and I believe it will be of broad impact in the chromatin and DNA repair fields. As such, I support publication. The only remaining comment is that I think the distinction between the structures in Fig. 5b,c could use a little clarification in the main text. My understanding is that these are derived from two distinct 3D classes of the same sample (per Ext. Data Fig. 7d) but this is somewhat unclear from the main text and figure legend. Otherwise, manuscript revisions are clear.

Referee #3:

In the revised manuscript all my comments and concerns have been addressed.

Related to the high radiation dose required to induce radio-sensitivity in yeast, the authors are reminded that the major determinant of this effect is the much reduced volume of the yeast nucleus in comparison to the mammalian nucleus. As such, the citation of the referenced literature (#27-30; line 232) should be carefully reconsidered and PMID:14162687 should be added.

Author Rebuttals to First Revision:

General comment)

The authors have addressed my concerns with additional experiments and revision of the text. The title and summary of the paper are well supported by the structures, mutagenesis, in vitro binding and yeast-based repair assays. The main take-home message that the N-terminal lobe domain (NLD) of RAD51 binds to nucleosome and helps to disassemble nucleosomes for DSB repair is clear and convincing.

With the structural models and maps provided by the authors, the following points may be considered to improve the accuracy and focus of the manuscript.

Reply)

Thank you very much for your favorable comment. We addressed your concerns as described below.

Comment 1)

As the nucleosome-bound octameric, nonameric, and decameric RAD51 rings share not a single conserved protein-protein or protein-DNA interface when the nucleosome or, a single subunit or NLD of RAD51 is superimposed among them. Fig. 2b-e appear unclear and yet redundant as no cryoEM map defines the K64 and K70 sidechains, and NLD containing these two conserved residues interacts differently with nucleosomes in each panel. The authors may simplify the figure and discussion by consolidating the structural findings and explain that K64 and K70 are chosen for mutagenic studies because they are conserved and have proximity to nucleosome DNA. Why is the conserved K73 not studied in the nucleosome binding and DSB repair?

Reply)

Thank you very much for this insightful comment. As this reviewer suggested, we explain that K64 and K70 were chosen for mutagenic studies because they are conserved and are close to the nucleosome DNA (p.6, ll.27-28). In addition, we also mention that K73 was not selected because it is farther away from the nucleosomal DNA, although it is conserved (p.6, ll.29-30).

Comment 2)

The structures of the Rad51-nucleosome complexes are of medium to low resolution. Even when overall structures appear 3.7–4.7 Å, which is largely due to the well-ordered nucleosome part, Rad51 and Rad51-nucleosome interface are often only at 5-6 Å. In addition to the limited resolutions of structures, these structures were obtained at a low and non-physiological salt concentration (30 mM NaCl), as were all binding studies. Structural interpretations need to take these facts into account. For example, the reported interaction between the histone H4 tail and NLD (Fig. 3) is not as convincing. In the EMSA assay, such complexes are only a small fraction of the “canonical” Rad51-nucleosome complexes independent of the H4 tail. The NLD in the structural model of H4 tail-NLD complex has very weak map volume, and its position is poorly defined and exhibits no contact with the histones. The H4 tail (K16 to and R19) does not contact NLD in the structural models, and instead they bind the nucleosomal DNA well.

Reply)

Thank you very much. The previous Fig. 3a was misleading, because the H4 tail did not fit well with the cryo-EM map. To visualize the H4 N-tail location more clearly, we re-modeled it and present the new Fig. 3a, in which the H4 N-tail angle fits well with the cryo-EM map. This new model structure has been deposited in the PDB and its validation report was submitted with the revised manuscript.

Comment 3)

As the structural data were obtained at 30 mM NaCl, dimerization of the RAD51 filament-nucleosome complexes shown in Fig. 6e is of concern. It does not support the main point of the manuscript, nor is it clear whether the dimerization is an artifact due to the low salt condition or functionally meaningful.

Reply)

According to this reviewer’s suggestion, we removed the text and Fig. 6e regarding the dimerization of the RAD51 filament-nucleosome complexes.

Other points for potential improvement:

1. Each figure can use more labels. For example, panels in Figs. 1 and 2 can use labels, such as octameric, nonameric, or decameric RAD51, and +ADP, +AMPPNP, +ATP or no nucleotide cofactor, etc. In Fig. 4, adding labels of “+AMPPNP”, “+ADP” to panels a,b and c,d, respectively, will help.

Reply)

We corrected them accordingly.

2. Please label complex names in all cryoEM work-flow ED figures. Each figure contains several different complexes. It is not easy to figure out which is what.

Reply)

We corrected them accordingly.

3. Is the map resolution 2.89 Å (ED. Fig. 3 and Table 1) a typo or overestimated for the structure of histone H4 tail-NLD complex (complex #7)? The model resolution is reported to be 4.79Å (ED Table 1), nearly 2 Å worse than the map resolution. The best part of the map (histones) appears no better than 3.5 Å, and the NLD is below 5 Å.

Reply)

Thank you very much. We mistakenly used the FSC threshold of 0.143 for the model resolution estimation. This value should be used for the map resolution, and the model resolution should be estimated with an FSC threshold of 0.5. We corrected these values in ED Table 1. The 2.89 Å value (ED Fig. 3) is the correct map resolution, but the representative 3D map in the same figure was ad-hoc low-pass filtered at 5.4 Å, so the RAD51 and nucleosome maps could both be clearly visualized. We explain this fact in the corresponding figure legends.

4. The labels of the y-axis in Fig. 3b,c contain a typo, “conformation” missing “t”. The quantification shown in the plots appears to quantify the complex of two RAD51 rings bound to one nucleosome and not the 1:1 complex, which is 100% and dominant. The legend and the plot label should clarify this.

Reply)

Thank you very much. We corrected Fig. 3b,c accordingly.

5. It is odd that the map resolutions of #5 (RAD51 filament bound to the nucleosome) and #9 complex are 6.77 and 7.81 Å, respectively, and the resulting structural models have the resolution of 4.90 and 4.79 Å. How could the models be so much better than the experimental data (maps)?

Reply)

Thank you very much. We mistakenly used the FSC threshold of 0.143 for the model resolution estimation. This value should be used for the map resolution, and the model resolution should be estimated with an FSC threshold of 0.5. We corrected these values in ED Table 1.

6. Resolutions of cryoEM maps and structures do not need double decimals as they are not so precisely estimated.

Reply)

We corrected these values, accordingly.

Referee #2:

Comment)

The revisions added substantially to the clarity of the manuscript and the functional relevance of the structures. All of my comments and queries were addressed. The study overall is incredibly exciting, and I believe it will be of broad impact in the chromatin and DNA repair fields. As such, I support publication. The only remaining comment is that I think the distinction between the structures in Fig. 5b,c could use a little clarification in the main text. My understanding is that these are derived from two distinct 3D classes of the same sample (per Ext. Data Fig. 7d) but this is somewhat unclear from the main text and figure legend. Otherwise, manuscript revisions are clear.

Reply)

Thank you very much. We described this fact as “The structures in panels **b** and **c** are derived from two distinct 3D classes of the same sample (Ext. Data Fig. 7d)”, in the Fig. 5 legend.

Referee #3:

In the revised manuscript all my comments and concerns have been addressed.

Related to the high radiation dose required to induce radio-sensitivity in yeast, the authors are reminded that the major determinant of this effect is the much reduced volume of the yeast nucleus in comparison to the mammalian nucleus. As such, the

citation of the referenced literature (#27-30; line 232) should be carefully reconsidered and PMID:14162687 should be added.

Reply)

Thank you very much. The paper suggested by this reviewer seems to be the best for comparing the radiation sensitivities between yeast and mammalian cells. Therefore, in the revised manuscript, we cited this suggested paper (PMID:14162687) instead of the previous #27-30.